# Mouse and human antibodies bind HLA-E-leader peptide complexes and enhance NK cell cytotoxicity

Dapeng Li [1,2,8], Simon Brackenridge [3,8], Lucy C. Walters[3,8], Olivia Swanson [1], Karl Harlos [4], Daniel Rozbesky[4,5], Derek W. Cain[1,2], Kevin Wiehe[1,2], Richard M. Scearce[1], Maggie Barr[1], Zekun Mu [1], Robert Parks[1], Max Quastel [3], Robert J. Edwards [1,2], Yunfei Wang[1,2], Wes Rountree[1,2], Kevin O. Saunders [1,6,7], Guido Ferrari[7], Persephone Borrow [3], E. Yvonne Jones [4], S. Munir Alam[1,2], Mihai L. Azoitei [1,2✉], Geraldine M. Gillespie [3✉], Andrew J. McMichael [3✉] & Barton F. Haynes [1,6✉]

The non-classical class Ib molecule human leukocyte antigen E (HLA-E) has limited polymorphism and can bind HLA class Ia leader peptides (VL9). HLA-E-VL9 complexes interact with the natural killer (NK) cell receptors NKG2A-C/CD94 and regulate NK cell-mediated cytotoxicity. Here we report the isolation of 3H4, a murine HLA-E-VL9-specific IgM antibody that enhances killing of HLA-E-VL9-expressing cells by an NKG2A$^+$ NK cell line. Structural analysis reveal that 3H4 acts by preventing CD94/NKG2A docking on HLA-E-VL9. Upon in vitro maturation, an affinity-optimized IgG form of 3H4 showes enhanced NK killing of HLA-E-VL9-expressing cells. HLA-E-VL9-specific IgM antibodies similar in function to 3H4 are also isolated from naïve B cells of cytomegalovirus (CMV)-negative, healthy humans. Thus, HLA-E-VL9-targeting mouse and human antibodies isolated from the naïve B cell antibody pool have the capacity to enhance NK cell cytotoxicity.

[1] Duke Human Vaccine Institute, Duke University School of Medicine, Durham, NC 27710, USA. [2] Department of Medicine, Duke University School of Medicine, Durham, NC 27710, USA. [3] Nuffield Department of Clinical Medicine, University of Oxford, Oxford OX3 7FZ, UK. [4] Division of Structural Biology, Wellcome Centre for Human Genetics, University of Oxford, Oxford OX3 7BN, UK. [5] Department of Cell Biology, Charles University, Prague 12800, Czech Republic. [6] Department of Immunology, Duke University School of Medicine, Durham, NC 27710, USA. [7] Department of Surgery, Duke University School of Medicine, Durham, NC 27710, USA. [8] These authors contributed equally: Dapeng Li, Simon Brackenridge, Lucy C. Walters. ✉email: mihai.azoitei@duke.edu; geraldine.gillespie@ndm.ox.ac.uk; andrew.mcmichael@ndm.ox.ac.uk; barton.haynes@duke.edu

N atural killer (NK) cells play critical roles in immune surveillance by discriminating normal from altered cells, and function by killing non-self malignant or pathogen-infected cells and by producing inflammatory cytokines[1–3]. Specific recognition of abnormal cells by NK cells relies on a series of activating and inhibitory receptors, including the killer immunoglobulin-like receptor (KIR) family and NKG2/CD94 heterodimeric receptors[3,4]. NK cell inhibitory receptors ligate human lymphocyte antigen (HLA) or major histocompatibility complex (MHC) class I molecules expressed on healthy cells as self. Conversely, cells lacking MHC class I are recognized by NK cells as "missing-self" and are sensitive to NK cell-mediated killing[5,6]. In humans, KIRs recognize classical HLA class Ia molecules[7–9], whereas the inhibitory NKG2A/CD94 heterodimeric receptor interacts with the non-classical HLA class Ib molecule HLA-E and is balanced by an activating receptor NKG2C/CD94[10–12]. While KIR expression is heterogeneous, NKG2A/CD94 is expressed on ~40% of human NK cells[9,13,14]. Unlike classical HLA class I molecules, HLA-E has limited polymorphism with only two expressed variants, HLA-E*01:01 and HLA-E*01:03, that differ only in residue 107, which is outside the peptide-binding groove[15]. The NKG2A/CD94/HLA-E pathway is considered to be an important immune checkpoint and has recently become a focus for NK cell-based immunotherapeutic strategies[4,16–19]. A subset of CD8+ T cells also express NKG2A/CD94, and inhibition of the NKG2A/CD94–HLA-E interaction has similar application in CD8+ T cell-based immunotherapy[4,20].

HLA-E engages with NKG2A/CD94 via a restricted subset of peptides VMAPRT(L/V) (V/L/I/F)L (designated VL9) that derive from the leader sequence of HLA-A, -C, -G and a third of HLA-B molecules[10,12,21,22]. HLA-E binds VL9 peptides, which stabilize HLA-E surface expression[10,12] on healthy host cells in which HLA-Ia expression is not perturbed and initiate recognition by NKG2A/CD94 or NKG2C/CD94 on NK cells. The binding affinity of HLA-E-VL9 peptide complexes for NKG2A/CD94 is greater than that for NKG2C/CD94, such that the inhibitory signal dominates to suppress aberrant NK cell-mediated cytotoxicity and cytokine production[12,23–26]. In addition, HLA-E and its murine or rhesus macaque homologs are capable of binding to a range of other host peptides and pathogen-derived peptides, including heat-shock protein 60 (Hsp60)-derived peptides[27], Mycobacterium tuberculosis (Mtb) peptides[28,29], and simian immunodeficiency virus (SIV) Gag peptides[30,31]. However, only VL9 peptide-loaded HLA-E can engage CD94/NKG2A and protect cells from NK cell cytotoxicity[27,32,33]. Hence, leader sequence VL9 peptides are essential not only for stabilizing HLA-E surface expression but also for mediating the role of HLA-E/NKG2A/CD94 in regulating NK cell self-recognition. However, it remains unclear if interruption of this pathway by specifically targeting HLA-E-peptide complexes on target cells can enhance NK cell activity.

Here, we isolated a murine IgM monoclonal antibody (mAb) 3H4 that bound to HLA-E-VL9 on target cells and enhanced NK cytotoxicity mediated by an NKG2A+ NK cell line. Crystallographic analysis of an HLA-E-VL9/3H4 antigen-binding fragment (Fab) co-complex indicated that, due to steric clashes, 3H4 and CD94/NKG2A cannot simultaneously bind to these overlapping recognition surfaces on HLA-E-VL9. Surprisingly, the Ig V(D)J residues at the 3H4-HLA-E-VL9 binding interface were germline-encoded. While 3H4 mAb enhanced NK cytotoxicity as an IgM, the IgG form of the antibody did not enhance NK cytolytic activity. To address this, we developed 3H4 IgG variants with enhanced HLA-E-VL9 binding affinity by high-throughput screening of antibody libraries. Optimized 3H4 IgG Abs contained mutations in their CDR-H3 loops, bound HLA-E/VL-9 ~220 times tighter than the WT mAb and showed robust enhancement of NK cytotoxicity.

Finally, human HLA-E-VL9-reactive, near-germline IgMs were isolated from the human naive B-cell repertoire that also enhanced NK cell killing as IgG. Thus, we identified a group of near-germline HLA-E-VL9-targeting antibodies in mice and male CMV seronegative humans that could regulate NK cell function in vitro.

## Results

**Isolation of murine HLA-E-VL9-specific mAb 3H4.** With the original intention of raising monoclonal antibodies to the HIV-1 Gag peptide RMYSPTSIL (RL9HIV) (the HIV counterpart of RL9SIV, one of the MHC-E binding SIV Gag epitope peptides identified previously[30]), we immunized human HLA-B27/β2-microglobulin (β2M) transgenic mice[34] (Supplementary Fig. 1a, b) with 293T cells transfected with surface-expressed single-chain HLA-E-RL9HIV complexes[35] (Supplementary Fig. 1c, d). We produced hybridomas, and screened culture supernatants for binding on a panel of 293T cells transfected with either single-chain HLA-E-RL9HIV peptide complexes, or with single-chain HLA-E-VL9 peptide complexes as a control. Unexpectedly, we isolated a subset of antibodies that specifically reacted with HLA-E-VL9 peptide, the most potent of which was the IgM mAb 3H4. Unlike the well-characterized pan-HLA-E mAb 3D12[36], 3H4 reacted specifically with HLA-E-VL9 (VMAPRTLVL) and not with control, non-VL9 HLA-E-peptide complexes (Fig. 1a). Mab 3H4 also bound to VL9 peptide-pulsed HLA-class I-negative K562 cells transfected with HLA-E[37] (Fig. 1b) and also to soluble HLA-E refolded with synthetic VL9 peptide in both enzyme-linked immunosorbent assay (ELISA) (Fig. 1c) and surface plasmon resonance (SPR) assays (Fig. 1d). SPR measurents showed that the 1:1 dissociation constants ($K_D$s) of IgM 3H4 and human IgG1 backbone 3H4 for soluble HLA-E-VL9 were 8.1 and 49.8 μM, respectively (Supplementary Fig. 1e).

Sequence analysis of 3H4 mAb revealed 1.04% heavy-chain variable region ($V_H$) and 2.51% light-chain viable region ($V_L$) mutations (Supplementary Data 1). We isolated three additional HLA-E-VL9 mouse mAbs from two additional immunization studies in mice (see Methods), and each of the four HLA-E-VL9 antibodies were minimally mutated IgMs (mean VH and VL mutations, 1.21% and 2.87%, respectively (Supplementary Data 1). Negative stain electron microscopy showed that the 3H4 IgM hybridoma antibody was predominantly pentameric with a small proportion of hexamers (Supplementary Fig. 1f, g). In addition, 3H4 was not autoreactive in anti-nuclear antibody or clinical autoantibody assays (Supplementary Fig. 1h, i).

**3H4 IgM recognized the α1/α2 domain of HLA-E and N-terminus of the VL9 peptide.** To map the epitope on the HLA-E-VL9 complex recognized by 3H4, we tested 3H4 binding to VL9 peptide presented by HLA-E, the rhesus ortholog Mamu-E, as well as two HLA-E/Mamu-E hybrids—one with HLA-E α1/Mamu-E α2 (Hα1/Mα2), the other with Mamu-E α1/HLA-E α2 (Mα1/Hα2). 3H4 did not bind to Mamu-E/VL9 or Hα1/Mα2-VL9, and its staining of cells expressing Mα1/Hα2-VL9 was weak (Fig. 1e), suggesting that 3H4 recognition involved interaction with both α1 and α2 domains of HLA-E, and the epitope on α2 might be partially conserved between human and rhesus. 3H4 also did not cross-react with mouse ortholog Qa-1b (Supplementary Fig. 1j). Moreover, VL9 mutations indicated that position 1 (P1) of the peptide was important for 3H4 binding (Fig. 1f), with strong antibody recognition of VL9 peptide P1 variants with alanine, cysteine, isoleucine, serine, threonine, weak binding to histidine and proline substitutions, but no interaction with arginine, glutamate, glycine, lysine, methionine, asparagine,

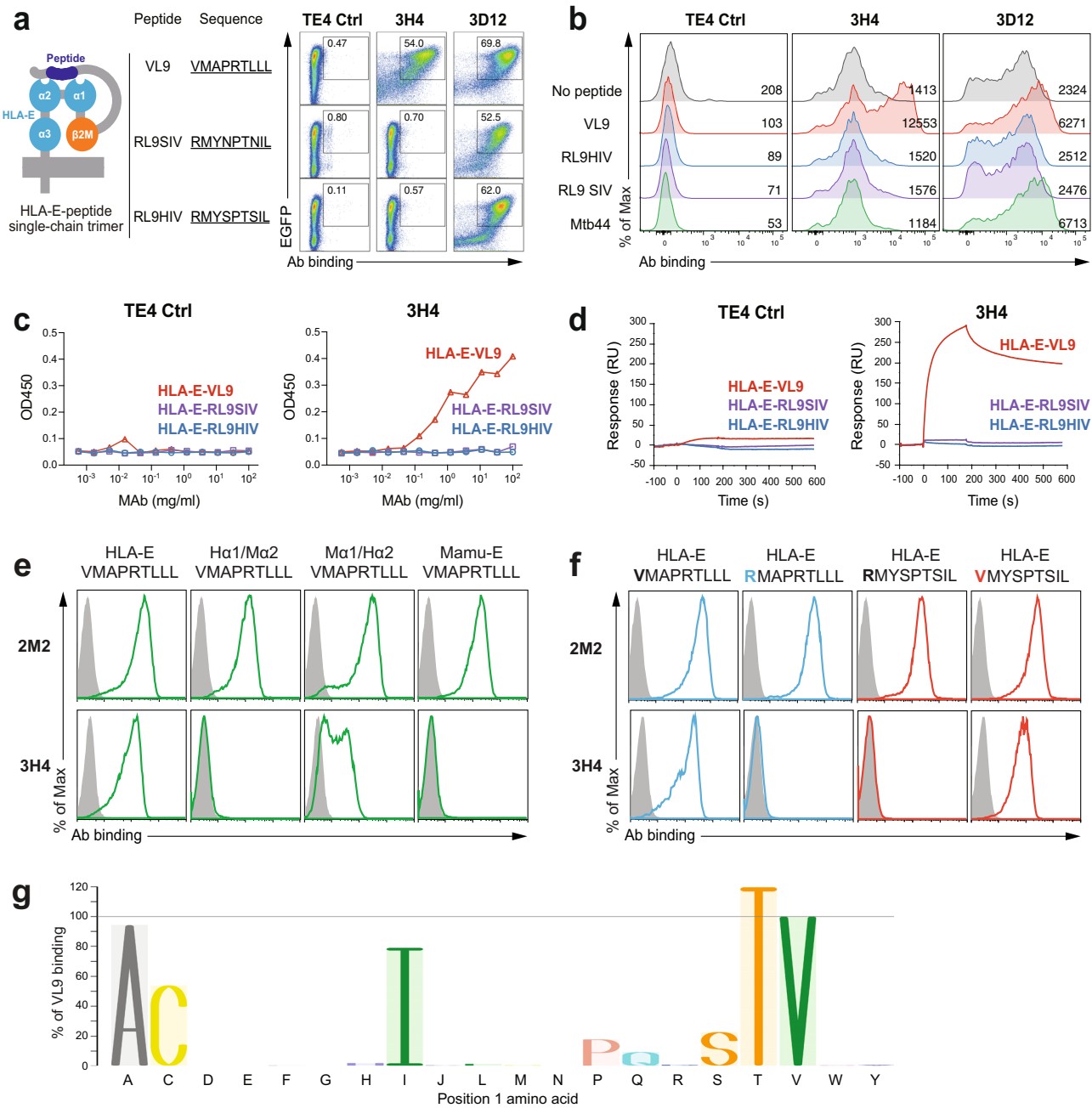

tryptophan, tyrosine or phenylalanine (Fig. 1g and Supplementary Fig. 1k). These data suggested that mAb 3H4 made contacts with both the HLA-E α1/α2 domain and the amino-terminal end of the VL9 peptide.

**Co-complex crystal structure of a 3H4 Fab bound to HLA-E-VL9.** A co-complex structure of 3H4 Fab bound to HLA-E-VL9, which packed in the C2 space group and diffracted to 1.8 Å (Table 1a), was obtained. One of the two copies present in the asymmetric unit is discussed here. 3H4 docked onto the N-terminal region of the HLA-E-peptide-binding groove making contacts with both the heavy-chain α-helices in addition to VL9 peptide residues 1–4 (Fig. 2a, b). The 3H4-HLA-E interface mainly comprised electrostatic interactions and was dominated by the 3H4 VH chain, which created a total buried surface area (BSA) of 1109.4 Å² and formed ten hydrogen bonds (H-bonds) and three salt bridges with HLA-E α1-helix residues and one H-bond with T163 of the HLA-E

α2-helix. The smaller 3H4 VL chain–HLA-E interface buried 522.8 Å² and involved only three inter-molecular H-bonds and three salt bridges (Table 1b–h). Superposition of the 3H4-HLA-E-VL9 co-complex with a published HLA-E-bound CD94/NKG2A structure[25,38] revealed steric clashes between the VH and VL domains of 3H4 and the CD94 and NKG2A subdomains, respectively (Fig. 2c, d). Moreover, seven HLA-E heavy-chain residues (α1-helix positions 58, 59, 62, 63 and α2 helix positions 162, 163 and 167) are shared 3H4-HLA-E and CD94/NKG2A-HLA-E footprints (Fig. 2e, f). Consequently, simultaneous docking of these two HLA-E binding partners, 3H4 and NKG2A/CD94, would likely be disallowed.

All four 3H4-derived residues that interfaced with the VL9 peptide (Y97, S100, S100A and Y100B) resided within the VH CDR3 D-junction and were germline-encoded. This 3H4-VL9 interface was characterized by weak Van der Waals and hydrophobic contacts, for example, Y100B (3H4) and V1 or P4

**Fig. 1 Isolation of monoclonal antibody 3H4 specifically targeting HLA-E-VL9 complex. a** 3H4 bound HLA-E-VL9 single-chain trimer (SCT)-transfected 293T cells. All SCT constructs express EGFP to indicate transfection efficiency. Transfected cells were stained with test antibody and then an Alexa fluor 555 (AF555)-anti-mouse Ig(H + L) secondary antibody. A control mouse IgM TE4 was used as a negative control. Anti-pan-HLA-E antibody 3D12 was used as a positive control. Representative data from one of five independent experiments are shown. **b** 3H4 bound VL9 peptide-pulsed K562-HLA-E cells. RL9HIV, RL9SIV, Mtb44 peptides served as peptide controls. TE4 and 3D12 were used as antibody controls. Peptide-pulsed cells were stained with test antibody and then an Alexa fluor 647 (AF647)-anti-mouse Ig(H + L) secondary antibody. Mean fluorescence intensity (MFI) of each sample is shown. Representative data from one of three independent experiments are shown. **c**, **d** 3H4 specifically bound to soluble HLA-E-VL9 complexes as measured by ELISA and SPR. **c** ELISA plates were coated with 3H4 or control IgM TE4 in serial dilution, blocked, and incubated with C-trap-stabilized HLA-E-VL9, HLA-E-RL9HIV, HLA-E-RL9SIV antigens. After washing, antigen binding was detected by adding HRP-conjugated anti-human β2M antibody. **d** For SPR, biotinylated HLA-E-peptide complexes (HLA-E-VL9, HLA-E-RL9SIV, HLA-E-RL9HIV and mock control) were bound to the immobilized streptavidin. Antibody 3H4 and control TE4 were flowed over sensor chips and antibody binding was monitored in real-time. Representative data from one of two independent experiments are shown. **e** 3H4 recognized the α2 domain of HLA-E. Flow cytometry analysis of 3H4 and 2M2 (a control β2M mAb) binding to 293T cells transfected with VL9 presented by HLA-E, Mamu-E, and two HLA-E/Mamu-E hybrids—one with HLA-E α1/Mamu-E α2 (Hα1/Mα2), the other with Mamu-E α1/HLA-E α2 (Mα1/Hα2) (green). Transfected cells were stained with test antibody and then an AF647-anti-mouse Ig(H + L) secondary antibody. Isotype control stained cells were used as negative controls (gray filled histograms). Representative data from one of three independent experiments are shown. **f** 3H4 recognized position 1 (P1) of the VL9 peptide. 3H4 and 2M2 (a control β2M mAb) staining of 293T cells transfected with HLA-E-VL9 (VMAPRTLLL) or HLA-E-VL9 with a mutation at P1 (valine to arginine; RMAPRTLLL) (blue), and with HLA-E-RL9HIV (RMYSPTSIL) or HLA-E-RL9HIV with a mutation at P1 (arginine to valine; VMYSPTSIL) (red). Transfected cells were stained with test antibody and then an AF647-anti-mouse Ig(H + L) secondary antibody. Isotype control stained cells were used as negative controls (gray filled histograms). Representative data from one of three independent experiments are shown. **g** 3H4 recognized peptides with variants in P1. 293T cells were transfected with HLA-E SCTs with VL9 peptides with single amino acid mutations at P1, then stained with 3H4 antibody followed by AF647 conjugated anti-mouse IgG(H + L) secondary antibody. Cells were gated for EGFP-positive subsets. MFI of 3H4 staining on wildtype VL9 peptide was set as 100%, and the percentages for binding to mutants calculated as (MFI of 3H4 binding on each P1 variant)/(MFI of 3H4 binding on wildtype VL9) × 100%.

(VL9) (Fig. 2g). Further, positioning of the Y100B (3H4) side chain directly above V1 (VL9) in part explained preference for small side chains at this peptide position and the dramatic reductions in 3H4 binding to VL9 variants with larger H or F residues at position 1 (Fig. 1g). Unique shape complementarity also featured at this interface with the side chains of S100 and S100A (3H4) wrapping around the cyclic side chain of P4 (VL9).

The germline-encoded VH CDR3 D-junction residues that formed the 3H4-VL9 interface (Y97, S100, S100A and Y100B), also mediated key HLA-E heavy-chain contacts. The surface loop residues A93-V102 swept across the HLA-E-peptide-binding groove forming H-bonds with both the α1 and α2 helices; HLA-E α2 helix T163 formed an H-bond with S100 (3H4), and HLA-E α1-helix R62 formed two H-bonds with the Y100B (3H4) mainchain and an additional H-bond with the mainchain of S100A (3H4) (Fig. 2h). Y100B (3H4) was involved in multiple polar pi stacking interactions. Not only was the Y100B side chain sandwiched between R62 and W167 of the HLA-E α1 and α2 helices, respectively, R62 (HLA-E α1) was also positioned between the aromatic rings of the VH CDR3 domain Y100B and W100D residues.

Key contacts outside the germline-encoded CDR3 D-junction region were also formed at the 3H4 VH- and VL-HLA-E interfaces. For 3H4 HC, the VH CDR2 region (residues I51-T57) was positioned above the HLA-E α1-helix where numerous inter-molecular H-bonds were formed involving VH CDR2 residues G56 and N54 in addition to D50, Q61 and K64 of the VH framework region (Fig. 2h). Critically, HLA-E α1-helix R65 residue formed four H-bonds with the 3H4 VH and also mediated polar pi stacking interactions with W100D of the VH CDR3 loop. For 3H4 LC, D92 and E93 of the VL CDR3 loop H-bond with K170 of the HLA-E α2-helix and N30 of the VL CDR1 loop formed an H-bond with the α2-helix residue, E166, of HLA-E (Fig. 2i). Notably, the four key interfacing residues of the 3H4 VH CDR3 D-junction (Y97, S100, S100A and Y100B) were germline-encoded (Fig. 2j).

**3H4 IgM enhanced NK cell cytotoxicity against HLA-E-VL9-expressing target cells.** Given the suppressive role of the HLA-E-VL9/NKG2A/CD94 pathway in NK cell function, we tested

whether the binding of mAb 3H4 to HLA-E-VL9 could enhance NK cell killing of target cells (Fig. 3a). An NKG2A/CD94-positive, CD16/CD32/CD64-negative human NK cell line, NK92 (Supplementary Fig. 2a, b), exhibited significantly increased cytotoxicity against HLA-E-VL9-transfected 293T cells (Fig. 3b; $P < 0.0001$, mixed effects models) but not against non-HLA-E-expressing 293T cells (Fig. 3c) in the presence of 3H4 IgM compared to an isotype control IgM. In addition, we tested a combination of 3H4 with the NKG2A-specific murine antibody, Z199. While Z199 alone enhanced NK killing against HLA-E-VL9-expressing cells, no additional elevation of killing was observed with the combination of mAbs 3H4 and Z199 (Fig. 3d, e), suggesting that killing enhancement was maximal with either 3H4 or Z199 alone. These data demonstrated that HLA-E-VL9-specific IgM mAb 3H4 could enhance the killing capacity of NKG2A+ NK cells in vitro by binding to HLA-E-VL9 on target cells.

The majority of multimeric IgM is restricted to serum and lymph and does not penetrate well into tissues[39]. Thus, we constructed a recombinant 3H4 IgG in a human IgG1 backbone and tested it for ability to enhance NK92 cell killing of HLA-E-VL9 target cells. In contrast to 3H4 IgM, 3H4 IgG could not mediate enhancement of NK cell killing (Supplementary Fig. 3a). Thus, either the affinity of the 3H4 Fab on an IgG was too low ($K_D = 49.8$ μM; Supplementary Fig. 1e), or a multimeric antibody is needed for for efficient blocking of HLA-E-VL9 binding to NKG2A/CD94 to enhance of NK killing.

To distinguish between the need for higher affinity versus multimerization of the IgM antibody for enhanced NK killing activity, we developed and analyzed 3H4 antibody libraries using high-throughput screening on the surface of yeast (Fig. 4a). A library was built that contained ~1.1 million 3H4 scFv variants with amino acid diversity at sites that were determined by structural analysis to interact with HLA-E-VL9 (Fig. 4b). Seventeen total residues located in the CDR loops of 3H4 were randomized in groups of four based on their proximity, and all possible combinations of amino acids were sampled at these sites (Supplementary Fig. 3b). The resulting 3H4 scFv library was transformed into yeast and screened for three rounds by fluorescence-activated cell sorting (FACS) for binding to

**Table 1 Crystallographic data for the 3H4 Fab and VL9-bound HLA-E co-complex structure.**

**a**

| Data collection | 3H4 Fab-HLA-E (VL9) |
|---|---|
| Crystallization | |
| Condition | 0.1 M NA HEPES pH 7; 20% PEG 8000 |
| Resolution Å | 54.74–1.80 (1.864–1.80) |
| Space group | C2 |
| Cell dimensions | a = 190.47 |
| | b = 73.69 |
| | c = 159.75 |
| | α = 90.0° |
| | β = 101.62° |
| | γ = 90.0° |
| Solvent content [%] (copies in AU) | 55 (2) |
| Unique reflections | 196,943 (19,307) |
| Completeness [%] | 98.1 (96.8) |
| $R_{merge}$ [I] | 0.057 (1.289) |
| I/sigma | 15.0 (0.5) |
| Multiplicity | 6.9 (7.1) |
| $CC_{1/2}$ | 0.998 (0.724) |
| **Refinement** | |
| No. of non-hydrogen atoms | 14,072 |

**d VL9 peptide-Fab HC interface**

| Peptide | Fab HC |
|---|---|
| P1 Val [CB, CG1, CG2] | P97 Tyr (CDR3) [CZ, OH] |
| P2 Met [O] | P100 Ser (CDR3) [OG] |
| P4 Pro [C, O, CB, CG, CD] | P100A Ser (CDR3) [N, C, O, CB, OG] |
| P5 Arg [CB, CG, CD, NH1] | P100B Tyr (CDR3) [N, CA, CB, CG, CD1, CD2, CE1, CE2, CZ, OH] |

**e HLA-E HC-Fab HC interface**

| HLA-E HC | Fab HC |
|---|---|
| P56 Gly (α1) [CA] | P33 Asn (CDR1) [ND2] |
| P57 Ser (α1) [N, CB, OG] | P47 Trp [CZ3, CH2] |
| P58 Glu (α1) [CA, O, CB, CG, CD, OE1, OE2] | P50 Asp [OD1] |
| P59 Tyr (α1) [CA, O, CB, CG, CD2, CE2] | P52 Asn (CDR2) [CB, ND2] |
| P61 Asp (α1) [C, O, CB, CG, OD1, OD2] | P54 Asn (CDR2) [ND2] |

**f HLA-E HC-Fab LC interface**

| HLA-E HC | Fab LC |
|---|---|
| P55 Glu (α1) [OE1] | P1 Asp [OD1, OD2] |
| P58 Glu (α1) [C, O, CB, CG, CD, OE1, OE2] | P27 Gln (CDR1) [NE2] |
| P59 Tyr (α1) [N, CA, CB] | P28 Asp (CDR1) [O] |
| P62 Arg (α1) [CB, CG, CD, NE] | P30 Asn (CDR1) [CB, CG, ND2, OD1] |
| P162 Asp (α2) [C, CB, CG, OD2] | P32 Tyr (CDR1) [CE2, OH] |
| P163 Thr (α2) [CA, OG1] | P92 Asp (CDR3) [O, CG, OD1, OD2] |
| P166 Glu (α2) [CA, CB, CG, CD, OE1, OE2] | P93 Glu (CDR3) [CA, CB, CG, CD, OE1, OE2] |
| P167 Trp (α2) [NE1, CZ2, CZ3, CH2] | P94 Phe (CDR3) [C, O, CB, CG, CD1, CD2, CE1, CE2, CZ] |
| P170 Lys (α2) [CD, CE, NZ] | P95 Pro (CDR3) [N, CA, CB, CG, CD] |
| P174 Lys (α2) [CE, NZ] | |

**Table 1 (continued)**

**a — Data collection**

| Data collection | 3H4 Fab-HLA-E (VL9) |
|---|---|
| Crystallization | 20% PEG 8000 |
| $R_{factor}$ [%] | 0.1954 (0.4164) |
| $R_{free}$ [%] | 0.2323 (0.4304) |
| r.m.s.d. bonds [Å][a] | 0.005 |
| r.m.s. angles [deg] | 0.69 |
| Ramachandran statistics Favored [%] | 98.31 |
| Disallowed [%] | 0 |

**b — Total buried surface area of the interface (Å²)**

| Interface | Å² |
|---|---|
| Peptide-Fab HC (Chains P-G) | 242.6 |
| HLA-E HC-Fab HC (Chains A-G) | 1109.4 |
| HLA-E HC-Fab LC (Chains A-H) | 522.8 |

**c — 3H4 Fab-HLA-E (VL9)**

| | RMSD (Å) |
|---|---|
| Chains A & C (HLA-E heavy chain) | 0.95 |
| Chains A & C | 0.51 |

**d — VL9 peptide-Fab HC interface**

| Peptide | Fab HC |
|---|---|
| P62 Arg (α1) [N, CA, CB, CG, CD, NE, CZ, NH2] | P56 Gly (CDR2) [CA, C, O] |
| P63 Glu (α1) [OE1] | P57 Thr (CDR2) [N, CA, C, O] |
| P65 Arg (α1) [CB, CD, NE, CZ, NH1, NH2] | P58 Ile [CA, CG1, CG2, CD1] |
| P66 Ser (α1) [OG] | P59 Tyr [N, O] |
| P60 Asn | |
| P69 Asp (α1) | |
| P154 Glu (α2) [CA, CG, OD1, OD2] | P61 Gln [CA, CG, OD1] |
| P155 His (α2) [CA, O, ND1, CE1] | P64 Lys [N, CB, CD, NE2, OE1] |
| P158 Ala (α2) [CA, O, CB] | P64 Lys [CE, NZ] |
| P159 Tyr (α2) [N, CA, CB, CD1] | P97 Tyr (CDR3) [CE2, CZ, OH] |
| P162 Asp (α2) [CB, OD2] | P99 Gly (CDR3) [CA, C, O] |
| | P100 Ser (CDR3) [N, CA, CB, OG] |

**f/g — HLA-E HC-Fab LC interface — Hydrogen bonds**

| No. | HLA-E HC | Distance (Å) | Fab LC |
|---|---|---|---|
| 1 | P62 Arg [NE] (α1) | 2.97 | P100B Tyr [O] (CDR3) |
| 2 | P62 Arg [NH2] (α1) | 2.93 | P100A Ser [O] (CDR3) |
| 3 | P62 Arg [NH2] (α1) | 3.06 | P100B Tyr [O] (CDR3) |
| 4 | P65 Arg [NH2] (α1) | 2.74 | P50 Asp [OD1] |
| 5 | P65 Arg [NH1] (α1) | 3.47 | P50 Asp [O] |
| 6 | P65 Arg [NH2] (α1) | 2.77 | P50 Asp [OD1] |
| 7 | P65 Arg [NH2] (α1) | 3.11 | P56 Gly [O] (CDR2) |
| 8 | P163 Thr [OG1] (α2) | 2.45 | P100 Ser [OG] (CDR3) |
| 9 | P69 Asp [OD1] (α1) | 3.37 | P54 Asn [ND2] (CDR2) |
| 10 | P58 Glu [OE2] (α1) | 2.86 | P61 Gln [N] |
| 11 | P61 Asp [OD2] (α1) | 2.76 | P64 Lys [NZ] |

**Salt bridges — HLA-E HC-Fab HC interface**

| No. | HLA-E HC | Distance (Å) | Fab HC |
|---|---|---|---|
| 1 | P65 Arg [NH1] (α1) | 2.74 | P50 Asp [OD1] |
| 2 | P65 Arg [NH2] (α1) | 2.77 | P50 Asp [OD1] |
| 3 | P61 Asp [OD2] (α1) | 2.76 | P64 Lys [NZ] |

h — Hydrogen bonds

**Table 1 (continued)**

| a | Data collection | d | VL9 peptide-Fab HC interface | | f | HLA-E HC-Fab LC interface | | |
|---|---|---|---|---|---|---|---|---|
| | 3H4 Fab-HLA-E (VL9) | | | | | | | |
| | Crystallization | | Peptide | Fab HC | No. | HLA-E HC | Distance (Å) | Fab LC |
| (HLA-E heavy chain —no α3 domain) | 20% PEG 8000 | | P163 Thr (α2) [CB, CG2, OG1] | P100A Ser (CDR3) [C, O, CB, OG] | 1 | P170 Lys [NZ] (α2) | 2.80 | P92 Asp [OD2] (CDR3) |
| Chains B & D | | 0.82 | P167 Trp (α2) [CE2, NE1, CZ2, CH2] | P100B Tyr (CDR3) [CA, O, CB, CG, CD1, CD2, CE1, CE2, CZ, OH] | 2 | P170 Lys [NZ] (α2) | 2.69 | P93 Glu [OE2] (CDR3) |
| (β2-microglobulin) | | | | | 3 | P166 Glu [OE2] (α2) | 2.89 | P30 Asn [ND2] (CDR1) |
| Chains E & G | | 0.89 | | P100D Trp (CDR3) [CZ3, CH2] | | | | |
| (Fab heavy chain) | | | | | | **Salt bridges** | | |
| Chains F & H (Fab light chain) | | 0.36 | | | No. | HLA-E HC | Distance (Å) | Fab LC |
| | | | | | 1 | P170 Lys [NZ] (α2) | 3.96 | P92 Asp [OD1] (CDR3) |
| 3H4 Fab-HLA-E(VL9) and 1MHE from O'Callaghan et al. (1998) | RMSD | | | | 2 | P170 Lys [NZ] (α2) | 2.80 | P92 Asp [OD2] (CDR3) |
| Chains A & A | | 0.669 | | | 3 | P170 Lys [NZ] (α2) | 2.69 | P93 Glu [OE2] (CDR3) |

**a** Crystallographic data collection and refinement statistics. AS ammonium sulfate. ᵃr.m.s.d.: root mean-square deviation from ideal geometry. Statistics for outer shell indicated in parentheses. AU asymmetric unit. Rfree equals the R-factor against 5% of the data removed prior to refinement.

**b–f** Inter-chain RMSD and inter-molecular interfaces in the 3H4-HLA-E-VL9 structure. **b** Table detailing the total buried surface area of the interface in $Å^2$ between the 3H4 VH, VL, and the VL9-bound HLA-E complex. **c** Table of RMSD (root mean-square deviation) in Å between chains of the 3H4-HLA-E-VL9 co-complex structure. Two copies of the 3H4 Fab-HLA-E-VL9 co-complex were present in the asymmetric unit and thus RMSD between chains related by non-crystallographic symmetry was calculated via Cα atom pairwise alignment on the PDBePISA server. Average Cα atom RMSD following pairwise alignment is also reported for the HLA-E heavy chain (HC) of 1MHE (Chain A), a previously published non-receptor-bound HLA-E complex, and the HLA-E HC from the 3H4-HLA-E-VL9 structure reported here (Chain A). **d** Table listing residues involved in the interface between the 3H4 VH and the VL9 peptide. **e** Table of interacting residues of the 3H4 VH and HLA-E HC interface. **f** Table of interacting residues of the 3H4 VL and HLA-E HC interface.

**g, h** Hydrogen bonding and salt bridges in the 3H4-HLA-E-VL9 structure. Table of hydrogen bonds and salt bridges formed between the 3H4 heavy chain (HC) and the HLA-E HC (**g**) and the 3H4 light chain and the HLA-E HC (**h**). Hydrogen bonding cut-offs according to the PDBePISA default criteria. 3H4 chain numbering is according to the Kabat scheme whereby alternate insertion codes (letters after the residue number) are added to variable length regions of the antibody sequence. 3H4 residues within the CDRs are shaded green and labeled "CDR1/2/3". The position of HLA-E HC residues either on the α1 or α2 helix is also noted. Amino acid atom abbreviations: C—mainchain Carbon atom, O—mainchain Oxygen atom, N—mainchain Nitrogen atom, CA—α-Carbon atom, CB—β-Carbon atom, CD—δ-Carbon atom, CE—ε-Carbon atom, CG— γ-Carbon atom, CH—η-Carbon atom, CZ—ζ-Carbon atom, OD—δ-Oxygen atom, OE— ε-Oxygen atom, OG—γ-Oxygen atom, OH—η-Oxygen atom, NE—ε-Nitrogen atom, NH—η-Nitrogen atom, NZ—ζ-Nitrogen atom.

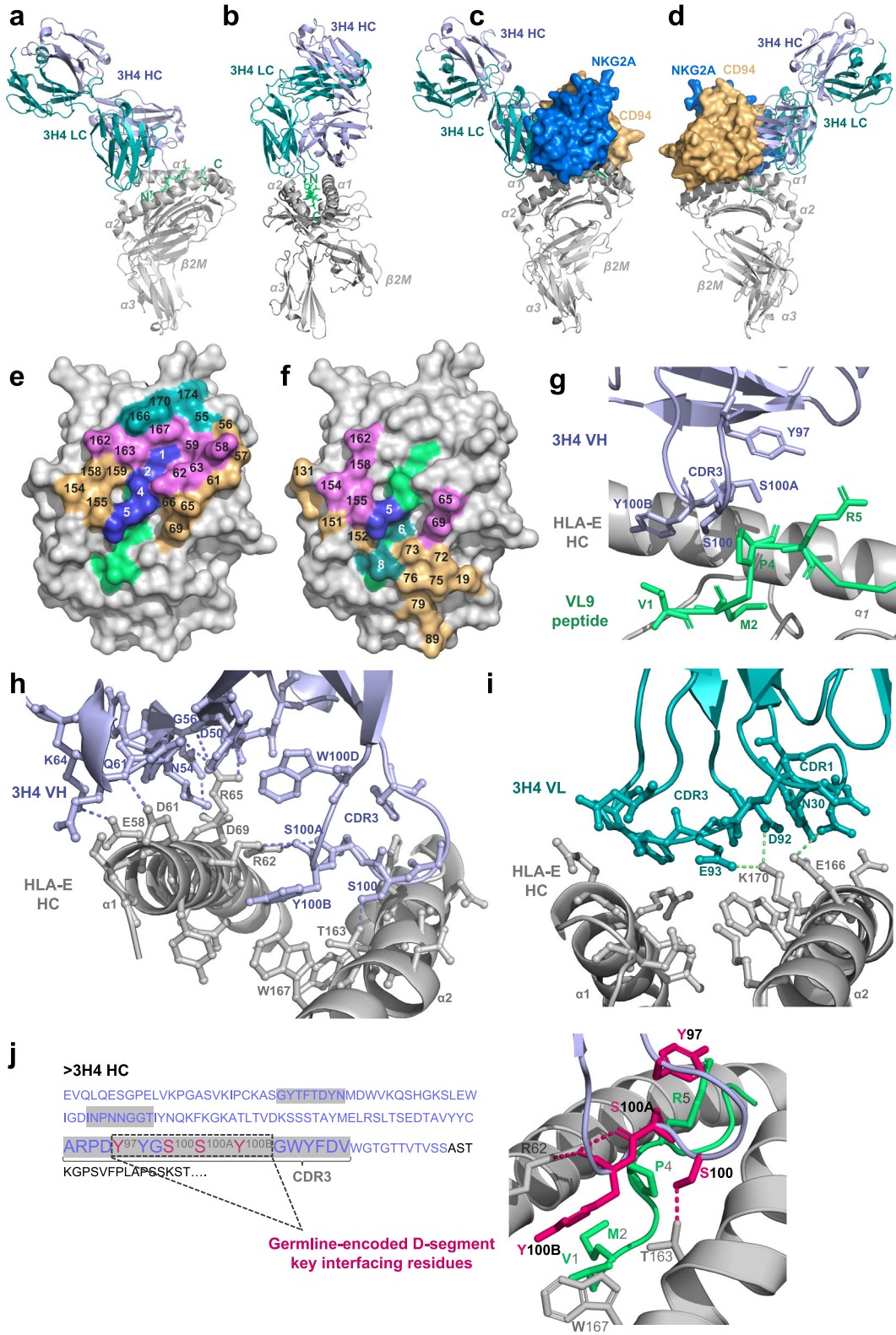

fluorescently labeled HLA-E-VL9 tetramer (Fig. 4c). Eleven 3H4 variants were selected for experimental characterization as recombinant human IgGs from the highly represented clones remaining in the library upon the final selection round. These novel Abs (3H4 Gv1 to 3H4 Gv12) were mutated at positions 97-100 of the CDR H3 loop. Compared to the original 3H4 mAb, the optimized antibodies predominantly contained small amino acids at positions 97 and 98, a polar amino acid at position 99, and a large aromatic at position 100 that is closest to the HLA-E-VL9 (Fig. 4d).

We next expressed all eleven 3H4 Gv antibodies recombinantnly as human IgGs, and confirmed that they had higher binding than wild-type 3H4 IgG on cell surface-expressed HLA-E-VL9 (Fig. 4e and Supplementary Fig. 3c). Two 3H4 variants, 3H4 G3v and 3H4 G6v, were selected for affinity and functional analysis. SPR measurments showed that the 1:1 dissociation

**Fig. 2 3H4 Fab-HLA-E-VL9 co-complex structural visualization. a, b** 3H4 Fab-HLA-E docking angles. HLA-E heavy chain and β2M light chain: gray; VL9 peptide: lime green; 3H4 heavy chain: light purple; 3H4 light chain: teal. **c, d** Superposition of 3H4 Fab and CD94/NKG2A docking sites on HLA-E. The HLA-E complex and 3H4 Fab are color-coded according to A and B. CD94: orange; NKG2A: marine blue. **e** Aerial view of the HLA-E-VL9 peptide-binding groove surface. Non-interfacing HLA-E residues: light gray; non-interfacing peptide residues: lime green; VL9 peptide residues involved in the 3H4 interface: marine blue. Interfacing HLA-E residues that contact 3H4 heavy and light chain: orange and teal, respectively; interfacing HLA-E residues that contact both 3H4 heavy and light chains: violet. Residue positions are numbered on the HLA-E surface view. **f** Aerial view of the overlapping 3H4 and CD94/NKG2A footprints on the HLA-E peptide-binding groove. VL9 peptide residues involved in both the 3H4 and CD94/NKG2A interfaces: marine blue; HLA-E heavy-chain residues involved in both interfaces: violet. Peptide and HLA-E heavy-chain residues involved exclusively in the CD94/NKG2A interface: teal and orange, respectively. **g** Binding interface of 3H4 HC/VL9 peptide. Interfacing residues (Y97, S100, S100A and Y100B of the VH CDR3 loop and V1, M2, P4 and R5 of the VL9 peptide) are shown in ball and stick-form with non-interfacing residues in cartoon form. VL9 peptide: lime green; HLA-E heavy chain: gray; 3H4 heavy chain: light purple. **h, i** Binding interfaces of 3H4 HC/HLA-E heavy chain (h) and 3H4 LC/HLA-E HC (i). Interfacing residues are displayed in ball-and-stick-form, non-interfacing residues are displayed in cartoon form and hydrogen bonds as dashed lines. **j** Key interfacing residues within the germline-encoded D-junction. 3H4 heavy-chain VH sequence were in purple and the CDR1/2/3 regions shaded gray. Germline-encoded residues within the VH CDR3 D-junction are denoted. The four key interfacing residues (Y97, S100, S100A and Y100B) within this germline-encoded D-junction that make contacts both the HLA-E heavy chain and VL9 peptide are highlighted magenta in the sequence and illustrated as magenta sticks in the PyMol visualization. HLA-E heavy chain: gray; VL9 peptide: green; hydrogen bonds: magenta dashed lines; residues of the 3H4 heavy chain that are not germline-encoded key interfacing residues: light purple.

---

constants ($K_D$s) of the selected 3H4 variants for soluble HLA-E-VL9 were markedly improved compared to that of wild-type 3H4 that had a $K_D$ of 49.8 μM. 3H4 G3v showed the tightest HLA-E-VL9 binding, with a $K_D$ of 220 nM, representing a ~226-fold improvement in affinity over the WT mAb (Fig. 4f). In the NK cytotoxicity assay, the optimized 3H4 mAbs enhanced NK-92 cell killing of HLA-E-VL9-transfected 293T cells at concentrations of 10 μg/ml and 1 μg/ml to levels comparable to those observed for 3H4 IgM (Fig. 4g and Supplementary Fig. 3d). Therefore, the higher affinity of affinity-optimized 3H4 IgG for HLA-E-VL9 could compensate for the need for avidity effect of 3H4 IgM multimers to mediate in vitro NK enhancement.

**Isolation of near-germline HLA-E-VL9-specific antibodies from CMV-negative, healthy humans.** We next asked if similar HLA-E-VL9 antibodies were present in the naive B-cell receptor (BCR) repertoire in humans, that could enhance NK killing of target cells. Using HLA-E-VL9 tetramers as probes, we identified B cells expressing HLA-E-VL9-specific B-cell receptors (BCRs) in four male, cytomegalovirus (CMV) seronegative human donors (Fig. 5a and Supplementary Fig. 4a, Supplementary Data 2). We isolated 56 HLA-E-VL9-specific antibodies that reacted with HLA-E-VL9 complexes but not control HLA-E-peptide complexes (Fig. 5b and Supplementary Fig. 4b, c Supplementary Data 3); all were IgM (Supplementary Data 3). By performing more in-depth analysis of the binding profiles of four representative HLA-E-VL9 antibodies—CA123, CA133, CA143 and CA147, we found that these antibodies exhibited differential cross-reactivities with rhesus Mamu-E-VL9 or mouse Qa-1-VL9 complexes (Supplementary Fig. 4d) in addition to distinct binding specificities to VL9 peptide variants (Supplementary Fig. 4e). The apparent affinities ($K_D$) of CA123 and CA147 on a human IgG1 backbone to soluble HLA-E-VL9 were 3.8 and 25.0 μM, respectively (Supplementary Fig. 4f). Human HLA-E-VL9 antibodies CA147 and CA123 tested in functional NK killing assays as a recombinant human IgG1. CA147 enhanced NK-92 cell cytotoxicity to HLA-E-VL9-expressing target cells (Fig. 5c), whereas CA123 had no enhancing effect (Supplementary Fig. 4g, h), suggesting that NK killing-enhancement function of the HLA-E-VL9 antibodies was determined by factors beyond binding affinity.

In the four humans, the percentages of HLA-E-VL9-specific B cells in pan-B cells (CD3⁻CD235⁻CD14⁻CD16⁻CD19⁺) were 0.0009%-0.0023% (mean of 0.0014%) (Fig. 5d). HLA-E-VL9-specific B cells were IgD⁺IgM⁺/⁻ B cells, in which four cell subsets were observed (Fig. 5e)—CD10⁻CD27⁻CD38⁺/⁻ naive B cells

(71.4%), CD10⁺CD27⁻CD38⁺⁺ immature or newly formed B cells[40] (10.7%), and CD10⁻CD27⁺CD38⁻ non-class-switched memory cells, demonstrating that BCRs targeting HLA-E-VL9 peptide existed in the naive B-cell repertoire of healthy humans.

**$V_H/V_L$ gene usage of HLA-E-VL9-specific antibodies.** To characterize the human antibody gene usage of HLA-E-VL9 antibodies, we analyzed the paired heavy-chain and light-chain gene sequences of 56 human HLA-E-VL9 antibodies, and found 1 multiple-member clone containing 6 antibodies in donor LP021[41] (Supplementary Data 3). Next, we compared the 51 HLA-E-VL9-specific B-cell clones with a reference human antibody repertoire[42]. Over 45% of the heavy-chain variable region ($V_H$) genes were VH3-21 or VH3-11 in HLA-E-VL9 antibodies, whereas less than 7% of the control B cells used these two genes (Fig. 5f and Supplementary Data 3). HLA-E-VL9 antibody light-chain variable regions ($V_κ/V_λ$) also were skewed and preferentially utilized IGKV3-15, IGKV1-39 and IGKV3-11 genes compared to controls (Fig. 5g and Supplementary Data 3). No J chain gene usage preference was observed (Supplementary Fig. 5a–d). Moreover, HLA-E-VL9 antibodies showed a trend to have shorter heavy-chain complementarity determining region 3 (CDR3) lengths than reference antibodies (Fig. 5h), while no difference was observed for light-chain CDR3 (Fig. 5i). Given that HLA-E-VL9 antibodies were IgMs derived primarily from naive or immature B cells, we compared the mutation frequencies of the 51 clones with a reference human antibody repertoire containing both naive and antigen-experienced antibodies[43]. Both HLA-E-VL9 antibody heavy and light-chain variable region genes exhibited low somatic mutation rates that were similar to naive B-cell controls (Fig. 5j, k). Thus, human HLA-E-VL9-specific antibodies were IgM, minimally mutated and displayed skewed usage of $V_H$ and $V_κ/V_λ$ genes.

**Discussion**

In this study, we have isolated and characterized antibodies reactive with HLA-E-VL9 peptide complexes, and found these antibodies were derived from the naive IgM B-cell BCR repertoire in mice as well as in HCMV seronegative male human blood donors. Somatic mutations of these antibodies were minimal, and the affinities of these antibodies for HLA-E-VL9 were low. The lack of class-switching in HLA-E-VL9-specific antibodies may reflect self-tolerance of CD4 T cells and a lack of T cell help for affinity maturation of these antibodies. While the mouse antibodies were selected in the setting of HLA-E-unrelated peptide

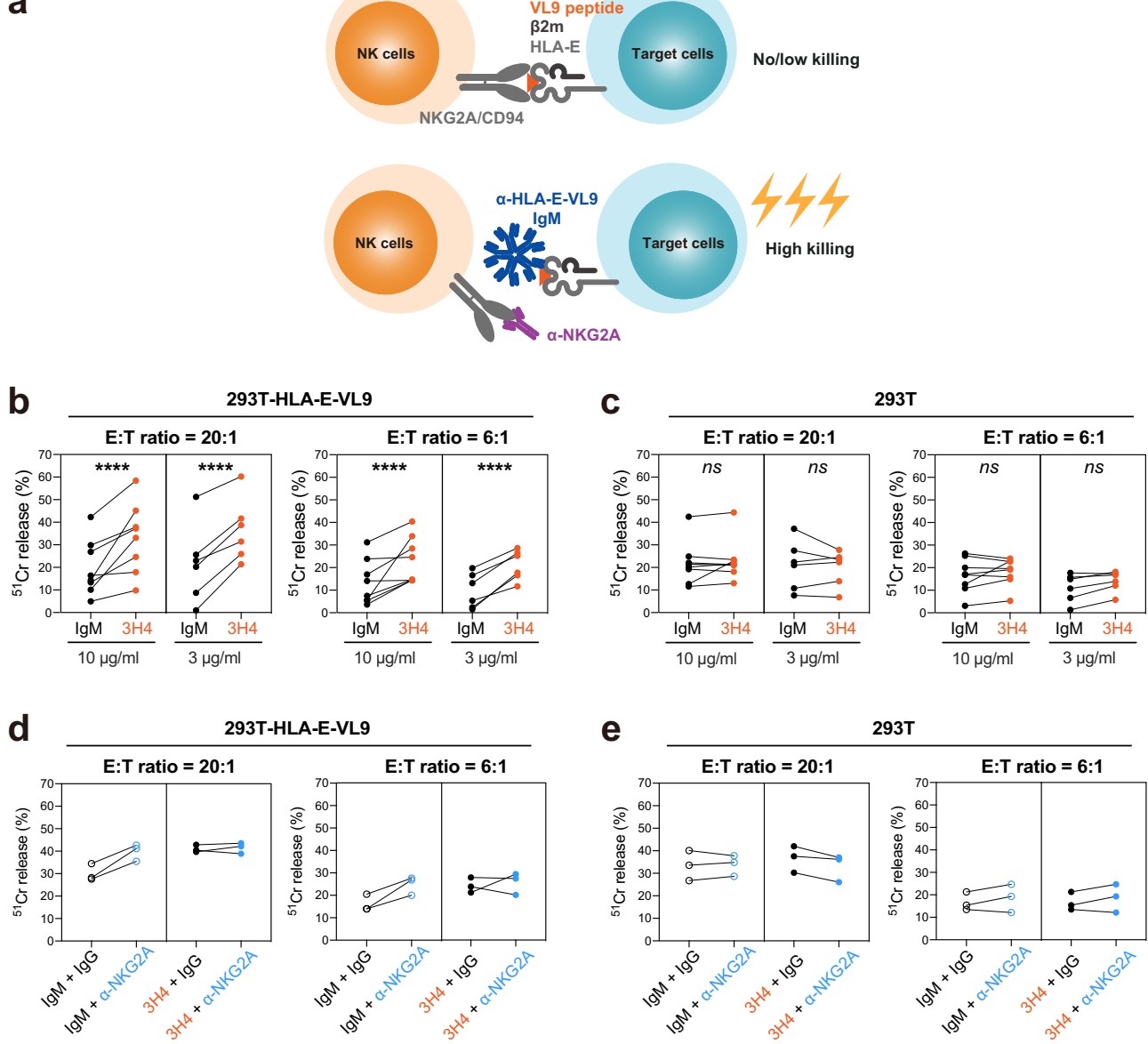

**Fig. 3 MAb 3H4 enhanced the cytotoxicity of the NKG2A + NK cell line NK-92 against HLA-E-VL9-expressing 293T cells. a** Schematic illustrating the hypothesis. Blockade of the inhibitory NKG2A/CD94/HLA-E pathway with anti-HLA-E-VL9 antibody (3H4) and/or anti-NKG2A antibody (Z199) could enhance target cell lysis by NK cells. **b**, **c** NK cell cytotoxicity against 3H4 IgM-treated target cells as assessed by $^{51}$Cr release assay. Antibody was incubated with HLA-E-VL9-transfected 293T cells (**b**) and untransfected 293T cells (**c**) at final concentration of 10 μg/ml or 3 μg/ml, and NK92 cells were added into the mixture as effector cells at effector: target (E:T) ratios of 20:1 and 6:1. Mouse IgM MM-30 was used as an isotype controls. Dots represent the mean values of triplicate wells in eight independent experiments. Statistical analysis was performed using mixed effects models. Asterisks show the statistical significance between indicated groups: ns, not significant, *$P < 0.05$, **$P < 0.01$, ***$P < 0.001$, ****$P < 0.0001$. **d**, **e** NK cell cytotoxicity in the presence of anti-NKG2A mouse IgG Z199 in combination with TE4 control- or 3H4-treated target cells as assessed by $^{51}$Cr release assay. Antibody combinations of Z199 + IgM control (**d**) or Z199 + 3H4 (**e**) were incubated with HLA-E-VL9-transfected 293T cells and untransfected 293T cells at a final concentration of 10 μg/ml, and NK92 cells were added into the mixture as effector cells. Dots represent the mean values of triplicate wells in three independent experiments.

immunizations, they were minimally mutated IgM antibodies, as were the antibodies isolated from human CMV-negative, healthy males. Structural analysis of the HLA-E-VL9-3H4 Fab co-complex revealed that the 3H4 heavy chain made key contacts with HLA-E and the VL9 peptide using germline-encoded residues in the CDR-H3 (D) region. However, 3H4 is a mouse antibody that reacted with human HLA-E-VL9. The HLA-E equivalent in C57BL/6xSJL mice is Qa1b, which presents a similar class Ia signal peptide AMAPRTLLL and 3H4 did not bind to this

HLA-E-peptide complex. Therefore, it remains unclear how 3H4 and possibly the other HLA-E-VL9-specific IgM antibodies were induced in the setting of HLA-E-unrelated peptide immunizations. In contrast, the human antibodies such as CA147 that bound to human HLA-E-VL9 were autoantibodies and were also near-germline.

Autoantibodies to HLA-Ia[44,45] and HLA-E heavy chains—have been detected in non-alloimmunized males, and contribute to allograft damage[48,49]. It has been suggested that the HLA-E

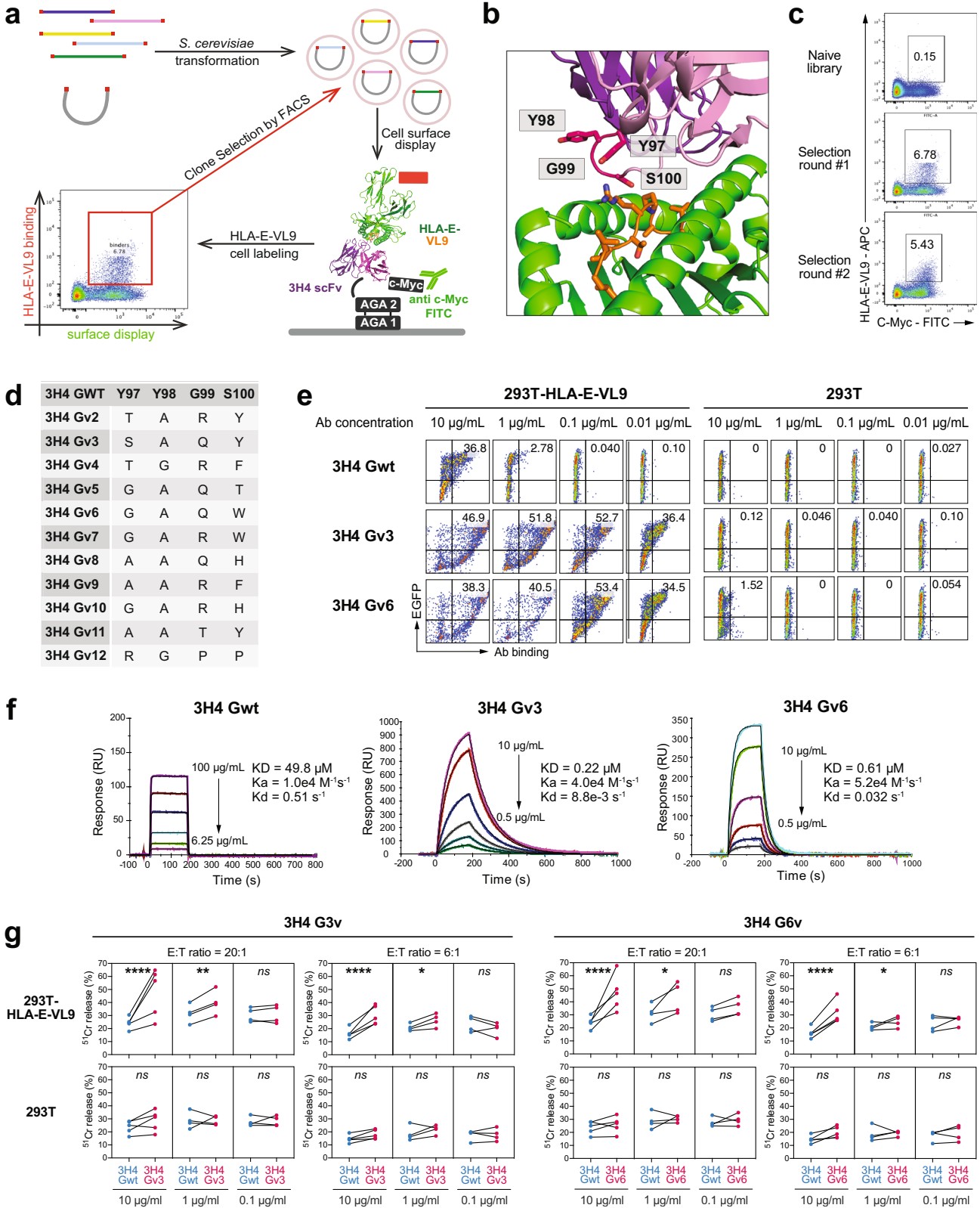

antibodies in non-alloimmunized humans could be elicited by autoantigens derived from soluble HLA-E heavy chains that become immunogenic without the β2M subunit, or viral/bacterial agents cross-reactive with HLAs[45–47,49]. It is of interest that human cytomegalovirus (CMV), which encodes the VL9 sequence VMAPRTLIL in the leader sequence of its UL40 gene. This peptide is processed in a TAP independent manner

and presented bound to HLA-E at the cell surface to inhibit NK cell killing and evade innate immune responses[50]. This has not been reported to elicit antibody responses, but HLA-E-UL40 peptide-specific T cells have been described when the limited polymorphism in the HLA A, B and C sequences mismatches that of the virally-encoded VL9 peptide sufficiently to overcome self-tolerance[51]. However, the subjects in our study were all HCMV

**Fig. 4 Affinity maturation of HLA-E-VL9-specific antibody 3H4 on human IgG1 backbone. a** Schematic illustration of the affinity maturation strategy. Libraries of 3H4 mAb variants were transformed into *S. cerevisiae* and displayed on the surface of yeast cells as single-chain fragment variable (scFv). APC-conjugated HLA-E-VL9 tetramers were used for FACS sorting. **b** Sites at the 3H4/HLA-E-VL9 interface where sequence optimization by library screening provideed the most significant affinity gains. 3H4: *purple*; HLA-E: *green*; VL9 peptide: *orange*. **c** Enrichment of HLA-E-VL9 + library clones after three rounds of selection by fluorescence-activated cell sorting (FACS). The yeast cells containing the scFv libraries were sorted sequentially for binding to decreasing concentrations of fluorescently labeled HLA-E-VL-9 (50 µg/ml, *top*; 10 µg/ml, *middle*; or 0.6 µg/ml *bottom*). **d** Mutations at positions 97-100 in the eleven 3H4 variants chosen for additional characterization upon library screening. **e** Binding of 3H4 Gwt and optimized variants to HLA-E-VL9 or HLA-E-Mtb44 transfected 293T cells. Representative flow cytometry data from one of three independent experiments are shown. **f** SPR sensorgrams showing binding kinetics of 3H4 Gwt and optimized variants. Rate constants ($k_a$, $k_d$) and dissociation constant $K_D$ were determined by curve fitting analysis of SPR data with a 1:1 binding model. Binding data are shown as colored lines, and the best fits of a 1:1 binding model are shown as black lines. Representative data from one of two independent experiments are shown. **g** Enhanced NK-92 cell cytotoxicity by optimized IgG 3H4 Gv3 and 3H4 Gv6 on HLA-E-VL9-transfected 293T cells and untransfected 293T cells, in compare with IgG 3H4 Gwt. Dots represent the mean values of triplicate wells in four or five independent $^{51}$Cr release assays. Statistical analysis was performed using mixed effects models. Asterisks show the statistical significance between indicated groups: ns, not significant, *$P < 0.05$, **$P < 0.01$, ***$P < 0.001$, ****$P < 0.0001$.

seronegative, ruling out the possibility that these antibodies were HCMV-induced. Similarly, that they were male, thus excluding pregnancy-induced priming. Therefore, while the origin and the physiological relevance of endogenous HLA-E-VL9 antibodies remain unclear, it is an intriguing possibility that HLA-E-VL9 antibodies may function to transiently mitigate NKG2A/CD94-mediated inhibition of NK cell cytotoxicity in vivo. Clearly, both mouse and human HLA-E-VL9 antibodies can enhance NK cytotoxicity in vitro, and their in vivo NK cell-enhancing capability warrents further exploration.

Finally, harnessing NK cells to attack tumor cells has emerged as an attractive strategy for cancer immunotherapies[52,53]. A promising target for therapeutic immune-modulation of NK cell functions is the inhibitory NKG2A/CD94-HLA-E-VL9 interaction. Monalizumab, the first-in-class monoclonal antibody checkpoint inhibitor targeting NKG2A, enhances anti-tumor immunity by activating cytotoxic activities of effector CD8+ T cells and NK cells[4,19,54]. In our study, co-complex structural analysis revealed steric clashes between the 3H4 Fab and the NK inhibitory receptor NKG2A/CD94 when docked onto HLA-E-VL9, thus explaining the mechanism of 3H4 IgM enhancing NKG2A+ NK cell killing. Notably, mouse 3H4 IgM, the affinity-optimized 3H4 IgG, and the recombinant IgG1 form of human CA147 each enhanced the cytotoxicity of an NKG2A + human NK cell line NK92, which is a safe and established cell line for adoptive immunotherapy in phase I clinical trials[55]. Thus, HLA-E-VL9-targeting antibodies 3H4 and CA147 could therefore potentially have therapeutic potential as NK checkpoint inhibitors.

## Methods

**Cell Lines**. K562-E cells (K562 cells stably expressing HLA-E) and K562-E/UL49.5 cells (with a TAP-inhibitor UL49.5) are kindly provided by Dr. Thorbald van Hall from Leiden University[37]. All the other cells used in this study are from ATCC. 293T cells (ATCC CRL-3216) were maintained in Dulbecco's Modified Eagle's Medium (DMEM; Gibco, Catalog# 10564) supplemented with 10% fetal bovine serum (FBS; Gibco, Catalog# 10099141) and 1% penicillin/streptomycin (Gibco, Catalog# 10378016). K562 cells (ATCC CCL-243), K562-E cells and K562-E/UL49.5 cells were cultured in Iscove's modified Dulbecco's Medium (IMDM; Hyclone, Catalog# SH30228.01) supplemented with 10% FBS. Jurkat, DU-4475 and U-937 cells were cultured in RPMI-1640 medium (Gibco, Catalog# 72400) supplemented with 10% FBS. SiHa cells were cultured in Minimum Essential Medium (MEM; Gibco, Catalog# 11095080) supplemented with 10% FBS. The NK-92 human cell line (ATCC CRL-2407) was cultured in Alpha Minimum Essential medium (α-MEM; Gibco, Catalog# 12561072) supplemented with 2 mM L-glutamine, 0.2 mM inositol, 0.1 mM 2-mercaptoethanol, 0.02 mM folic acid, 100 U/ml recombinant IL-2 (Biolegend, Catalog# 589108), 12.5% horse serum (Gibco, Catalog# 16050122) and 12.5% FBS. All the cells were maintained at 37 °C, 5% $CO_2$ in humidified incubators.

**Animals**. Transgenic mice carrying human β2-microglobulin (β2m) and HLA-B*27:05 genes were obtained from Jackson lab (B6.Cg-Tg(B2M,HLA-B*27:05)56-

3Trg/DcrJ; stock# 003428). Hemizygous mice were used in this experiment, as this strain is homozygous lethal. For hemizygous mice genotyping, peripheral blood lymphocytes (PBLs) were isolated and stained using mouse CD45 antibody (Biolegend, Catalog# 103122), human HLA class I antibody (Biolegend, Catalog# 311406) and human β2m antibody (Biolegend, Catalog# 316312). All animal experiments were conducted with approved protocols from the Duke University Institutional Animal Care and Use Committee.

**Human subjects**. Human leukapheresis frozen vials were collected by the External Quality Assurance Program Oversight Laboratory (EQAPOL)[56,57]. Samples from four male donors were used in this study. Supplementary Data 2 shows the clinical characteristics of the individuals studied. All experiments that related to human subjects was carried out with the informed consent of trial participants and in compliance with Institutional Review Board protocols approved by Duke University Medical Center.

**Peptide synthesis**. The VL9 peptide (VMAPRTVLL) was synthesized to >85% purity via Fmoc (9-fluorenylmethoxy carbonyl) chemistry by Genscript USA and reconstituted to 200 mM in dimethyl sulfoxide (DMSO).

**HLA-E-peptide protein refolding and purification**. β2-microglobulin, previously purified from inclusion bodies in a Urea-MES buffer, was added to a refolding buffer to achieve a final concentration of 2 µM. The refold buffer comprised 100 mM Tris pH 8.0, 400 mM L-arginine monohydrochloride, 2 mM EDTA, 5 mM reduced glutathione and 0.5 mM oxidized Glutathione and was prepared in MiliQ water. A 20 µM concentration of VL9 peptide (VMAPRTVLL), previously reconstituted to 200 mM in DMSO, was added to the refolding buffer followed by HLA-E*0103 heavy chain, which was pulsed into the refold to a final concentration of 1 µM. Once the refold had incubated for 72 h at 4 °C it was filtered through a 1.0 µm cellular nitrate membrane and concentrated in the VivaFlow 50R and VivaSpin Turbo Ultrafiltration centrifugal systems with 10 kDa molecular weight cutoffs. The concentrated samples were injected onto a Superdex S75 16/60 column and refolded protein eluted according to size into phosphate-buffered saline (PBS). Eluted protein complexes were validated by non-reducing sodium dodecyl-sulfate polyacrylamide gel electrophoresis (SDS-PAGE) electrophoresis on NuPAGE 12% Bis-Tris protein gels and further concentrated via VivaSpin Turbo Ultrafiltration centrifugal device to 1.1 mg/ml.

**HLA-E-peptide biotinylation and tetramer generation**. HLA-E-peptide samples requiring biotinylation were subsequently buffered exchanged on Sephadex G-25 PD10 columns (GE Healthcare, UK) into 10 mM Tris buffer using commercially available BirA enzyme (Avidity, USA) following the manufacturer's instructions. Following overnight biotinylation, protein samples were subsequently purified into 20 mM Tris pH 8, 100 mM NaCl buffer or PBS on a HiLoad 16/600 Superdex 75 pg column using an AKTA size-exclusion fast protein liquid chromatography (FPLC) system. Correctly folded β2m-HLA-E*01:03-peptide complexes were subsequently concentrated to 2 mg/ml and snap frozen.

HLA-E*01:03 tetramers were generated via conjugation to various fluorescent labels including Extravidin-PE (Sigma), Streptavidin-bound APC (Biolegend, San Diego) or BV421 (Biolegend, San Diego) at a Molar ratio of 4:1 as previously described[12].

**Immunization in HLA-B27/β2m transgenic mice**. HLA-B27/β2m transgenic mice ($n = 23$) were intramuscularly (i.m.) immunized with pooled HLA-E-RL9HIV complex (12.5 µg/animal) and HLA-E-RL9SIV complex (12.5 µg/animal) adjuvanted with STR8S-C[58] at weeks 0, 2, 4, 6, 12 and 16. MAb 3H4 was isolated from this study. In another experiment, HLA-B27/β2m transgenic mice ($n = 10$) were i.p. immunized with either HLA-E-RL9HIV single-chain trimer (SCT)

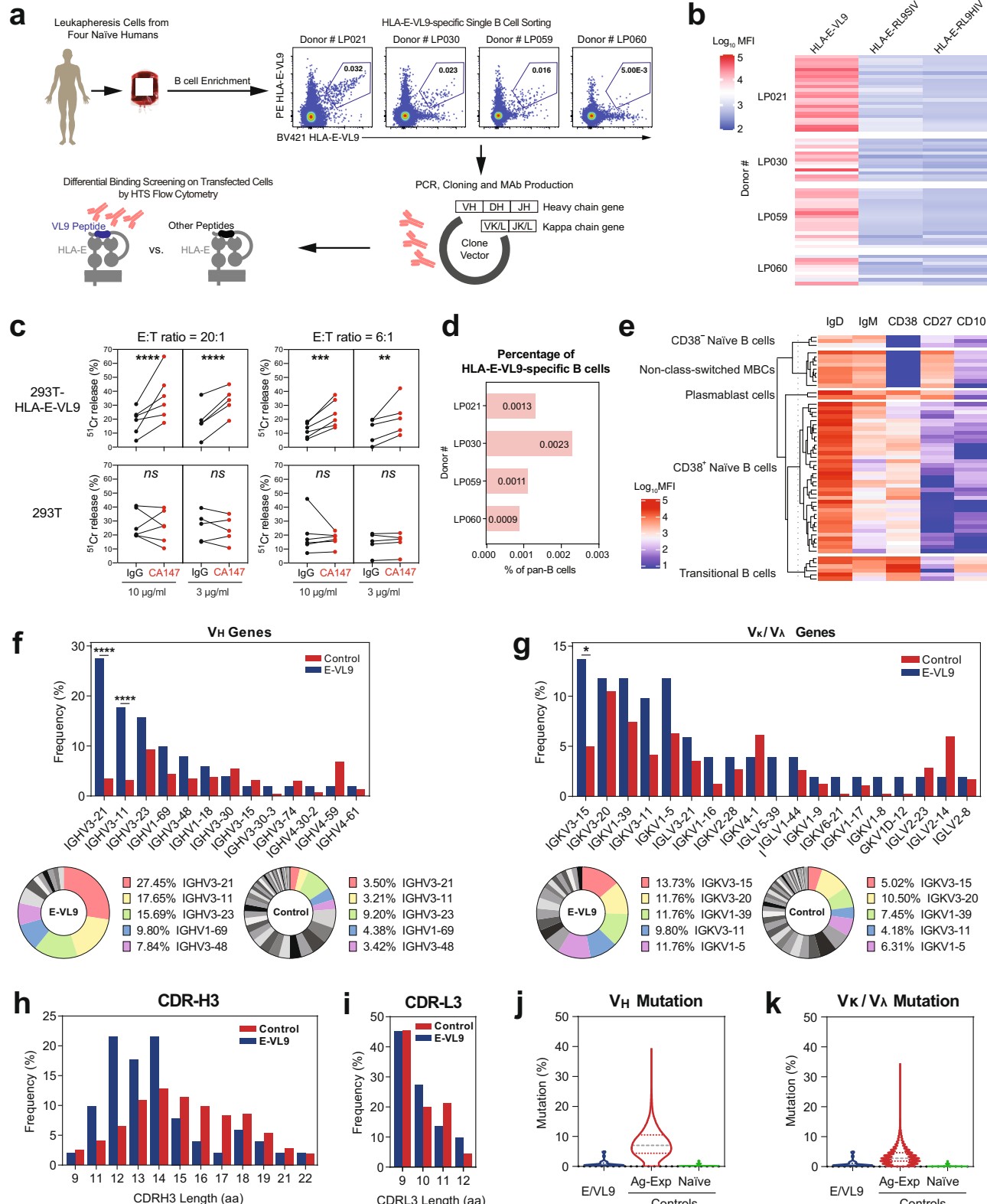

transfected 293T cells (2 x 10^6 cells/animal) or HLA-E-RL9SIV SCT transfected 293T cells (2 x 10^6 cells/animal) at weeks 0, 2, 4, 6, 17 and 19. MAb 13F11 was isolated from this study. In the third experiment, HLA-B27/β2m transgenic mice (n = 10) were i.m. immunized with HLA-E-VL9 complex (25 μg/animal) adjuvanted with STR8S-C at Week 0, 2 and 4, following by intraperitoneally (i.p.) immunization with HLA-E-VL9 SCT transfected 293T cells (2 x 10^6 cells/animal) at week 14, 16 and 18. MAb 10C10 and 2D6 were isolated from this study. Serum titers were monitored by ELISA Mice with high-binding antibody titers were selected for the subsequent spleen cell fusion and B-cell sorting experiments.

**Hybridoma cell line generation and monoclonal antibody production.** Mice were boosted with the indicated priming antigen 3 days prior to fusion. Spleen cells were harvested and fused with NS0 murine myeloma cells using PEG1500 to generate hybridomas. After 2 weeks, supernatant of hybridoma clones were collected and screened by flow cytometry-based high-throughput screening (HTS). Specifically, we tested for antibodies differentially binding 293T cells transiently transfected with plasmid DNA expressing single-chain peptide-HLA-E-ß2m trimers so that they expressed HLA-E-RL9HIV, HLA-E-RL9SIV or HLA-E-VL9 at the cell surface. Hybridomas cells that secreted HLA-E-VL9 antibodies were cloned

**Fig. 5 HLA-E-VL9-specific antibodies isolated from the B-cell pool of healthy humans. a** Scheme of isolating HLA-E-VL9-specific antibodies from healthy humans. Pan-B cells were first isolated by negative selection from human leukapheresis PBMCs. A three-color sorting strategy was used to sort single B cells that were positive for HLA-E-VL9 and negative for HLA-E-RL9HIV or HLA-E-RL9SIV. Flow cytometry data showing the sorting of HLA-E-VL9 double-positive, HLA-E-RL9HIV negative, HLA-E-RL9SIV negative B cells in PBMCs from four donors (LP021, LP030, LP059 and LP060) are shown. Variable regions of antibody heavy and light-chain genes were isolated from the sorted B cells by PCR, and cloned into an expression backbone with a human IgG1 constant region. Antibodies were produced by transient transfection in 293i cells, and antibody binding specificities were analyzed by surface staining of transfected 293T cells and high-throughput screening (HTS) flow cytometry. **b** Binding specificities of the HLA-E-VL9-specific antibodies ($n = 56$) from four donors shown as a heatmap. The compensated MFIs of HLA-E-VL9-specific antibody staining on HLA-E-VL9, HLA-E-RL9HIV, or HLA-E-RL9SIV transfected 293T cells at a concentration of 1 μg/ml were shown. Representative data from one of two independent experiments are shown. **c** NK cell cytotoxicity against CA147 IgG-treated target cells as assessed by $^{51}$Cr release assay. Human antibody CA147 was incubated with HLA-E-VL9-transfected 293T cells and untransfected 293T cells at final concentration of 10 μg/ml or 3 μg/ml, and NK92 cells were added into the mixture as effector cells at effector: target (E:T) ratio of 20:1 and 6:1. Human antibody A32 was used as the isotype control. Dots represent the mean values of triplicate wells in five independent experiments. Statistical analysis was performed using mixed effects models. Asterisks show the statistical significance between indicated groups: ns, not significant, *$P < 0.05$, **$P < 0.01$, ***$P < 0.001$, ****$P < 0.0001$. **d** Percentage of HLA-E-VL9-specific B cells in CD19$^+$ pan-B cells in four donors. **e** Phenotypes of HLA-E-VL9-specific B cells ($n = 56$) shown as heatmap. Expression of markers in each single B cell were determined from index sorting data and are shown as MFIs after compensation. Compensated MFIs below zero were set as zero. Each row indicates one single cell. The rows were clustered by K-means Clustering in R. Four subsets were observed: CD10$^-$CD27$^-$CD38$^{+/-}$ naive B cells, CD10$^+$CD27$^-$CD38$^{++}$ transitional B cells, CD10$^-$CD27$^+$CD38$^-$ non-class-switched memory B cells, and CD10$^-$CD27$^+$CD38$^+$ plasmablast cells. Detailed information for each single cell and antibody is shown in Supplementary Data 3. **f, g** Antibody gene usages. **f** Heavy chain viable ($V_H$) region gene usage shown as a bar chart (left) and pie chart (right). The top five $V_H$ genes found in HLA-E-VL9-specific antibodies are colored in the pie charts. **g** Kappa chain variable ($V_κ$) and lambda chain variable ($V_λ$) region gene usage shown as a bar chart (left) and pie chart (right). The top five $V_κ/V_λ$ genes found in HLA-E-VL9-specific antibodies are colored in the pie charts. Reference VH-VL repertoires ($n = 198,148$) from three healthy human donors from a previous study (*DeKosky, Nat Med 2015*) were used as a control. The chi-square test of independence was performed to test for an association between indicated gene usage and repertoire/antibody type in panels **a, b**. ****$P < 0.0001$; *$0.01 < P < 0.05$. **h, i** Comparison of heavy chain (**h**) and light chain (**i**) CDR3 (CDR-H3) length. HLA-E-VL9 antibody CDR-H3 length was compared with the ref. [42] human antibody CDR-H3 length. **j, k** Violin plots showing the mutation rates of heavy chains (**j**) and light chains (**k**). HLA-E-VL9 antibody sequences (E-VL9) were compared with reference sequences from naive and antigen-experienced (Ag-Exp) antibody repertoires ($n = 13,780$ and $34,692$, respectively).

by limiting dilution for at least five rounds until the phenotypes of all limiting dilution wells are identical. IgG mAbs were purified by protein G affinity chromatography, while IgM mAbs were purified by ammonium sulfate precipitation and by Superose 6 column size-exclusion chromatography in AKTA Fast Protein Liquid Chromatography (FPLC) system. The VH and VL sequences of mAbs were amplified from hybridoma cell RNA using primers reported previously[59,60].

**Cell-surface staining and high-throughput screening (HTS).** HLA-E SCT constructs encoding HLA-E-VL9, HLA-E-RL9HIV, or HLA-E-RL9SIV were transfected into 293T cells using GeneJuice transfection reagent (Novagen, Catalog# 70967). For epitope mapping experiment, a panel of HLA-E-VL9 SCT constructs with single amino acid mutations were transfected into 293T cells using the same method. Cells were dissociated with 0.1% EDTA at 48 h post-transfection and stained with a Fixable Near-IR Dead Cell Stain Kit (Thermo Fisher, Catalog# L34976). After washing, primary antibodies (supernatant from hybridoma cells, supernatant from transfected cells, or purified antibodies) were added and incubated with cells for 1 h at 4 °C, following by staining with 1:1000 diluted secondary antibodies for 30 min at 4 °C. For mouse primary antibodies, we used Alexa Fluor 555 (AF555) conjugated goat anti-mouse IgG (H + L) (Thermo Fisher, Catalog# A32727) or Alexa Fluor 647 (AF647) conjugated goat anti-mouse IgG (H + L) (Thermo Fisher, Catalog# A32728) as secondary antibodies; for human primary antibodies, we used AF555 conjugated goat anti-human IgG (H + L) (Thermo Fisher, Catalog# A-21433) or AF647 conjugated goat anti-human IgG (H + L) (Thermo Fisher, Catalog# A-21445) as secondary antibodies. Cells were then washed three times and resuspended in fixation buffer (1% formaldehyde in PBS, pH 7.4). Data were acquired on a BD LSR II flow cytometer and analyzed using FlowJo version 10.

**3H4 Fab production.** A humanized version of the 3H4 antibody (3H4-huIgG1) was digested to produce Fab fragments using the Pierce Fab Preparation kit (ThermoFisher SCIENTIFIC). 3H4 Fab-retrieved sample was further purified by size-exclusion on a Superdex S75 16/60 column and eluted into PBS buffer. Following concentration to 1.1 mg/ml and SDS-PAGE gel-based validation, 3H4 Fab purified material was incubated for 1 h on ice with freshly purified HLA-E-VL9. The combined 3H4:Fab-HLA-E-VL9 sample was concentrated to 7.5 mg/ml prior to crystallographic set-up.

**Crystallization screening.** Crystals were grown via sitting drop vapor-diffusion at 20 °C in a 200 nl drop with a 1:1 protein to reservoir ratio[61]. The 3H4 Fab-HLA-E(VL9) co-complex crystallized in 20% PEG 8000, 0.1 M Na HEPES at pH 7, in the ProPlex sparse matrix screen. Crystals were cryo-preserved in 25% glycerol and diffraction data were collected at the I03 beamline of Diamond Light Source.

**Crystallographic analysis.** Two copies of the co-complex structure of 3H4 Fab bound to HLA-E-VL9 were present in the asymmetric unit, a single copy constituted the focus of further discussion since root-mean-square deviation (RMSD) calculations from Cα-atom pairwise alignment of the two copies indicated minimal repositioning of interfacing residues at the HLA-E-3H4-binding site (Table 1b–f). Additionally, pairwise alignment with the previously published non-receptor-bound HLA-E coordinates (PDB ID: 1MHE)[62] revealed minimal structural changes in HLA-E upon 3H4 engagement (Table 1c).

Diffraction data were merged and indexed in xia2 dials[63]. Outer shell reflections were excluded from further analysis to ensure the $CC_{1/2}$ value exceeded the minimum threshold (>0.5) in each shell[64]. Sequential molecular replacement was carried out in MolRep of the CCP4i suite using molecule one of the previously published Mtb44-bound HLA-E structure with the peptide coordinates deleted (PDB ID: 6GH4) and one molecule of the previously published anti-APP-tag Fab structure (PDB ID: 6HGU) as phasing models[65,66]. Rigid body and retrained refinement were subsequently carried out by Phenix.refine[67] in between manual model building in Coot[68]. Model geometry was validated by MolProbity[69] and structural interpretation was conducted using the PyMOL Molecular Graphics System, version 2.0 (Schrödinger, LLC) in addition to the PDBePISA[70] and PDBeFOLD[71] servers.

**Antigen-specific single B-cell sorting.** HLA-E-VL9-specific human B cells were sorted in flow cytometry using a three-color sorting technique. Briefly, the stabilized HLA-E-β2M-peptide complexes were made as tetramers and conjugated with different fluorophores. Human pan-B cells, including naive and memory B cells, were isolated from PBMCs of healthy donors using human pan-B-cell enrichment kit (STEMCELL, Catalog# 19554). The isolated pan-B cells were then stained with IgM PerCp-Cy5.5 (Clone# G20-127, BD Biosciences, Catalog# 561285), IgD FITC (Clone# IA6-2, BD Biosciences, Catalog# 555778), CD3 PE-Cy5 (Clone# HIT3a, BD Biosciences, Catalog# 555341), CD235a PE-Cy5 (Clone# GA-R2, BD Biosciences, Catalog# 559944), CD10 PE-CF594 (Clone# HI10A, BD Biosciences, Catalog# 562396), CD27 PE-Cy7 (Clone# O323, eBioscience, Catalog# 25-0279), CD16 BV570 (Clone# 3G8, Biolegend, Catalog# 302035), CD14 BV605 (Clone# M5E2, Biolegend, Catalog# 301834), CD38 APC-AF700 (Clone# LS198-4-2, Beckman Coulter, Catalog# B23489), CD19 APC-Cy7 (Clone# LJ25C1, BD Biosciences, Catalog# 561743) and tetramers at 2 μg/million cells (including BV421-conjugated HLA-E-VL9 tetramer, PE-conjugated HLA-E-VL9 tetramer, APC-conjugated HLA-E-RL9SIV tetramer and APC-conjugated HLA-E-RL9HIV tetramer). The cells were then stained with a Fixable Aqua Dead Cell Stain Kit (Invitrogen, Catalog# L34957). HLA-E-VL9-specific B cells were sorted in BD FACSAria II flow cytometer (BD Biosciences) for viable CD3$^{neg}$/CD14$^{neg}$/CD16$^{neg}$/CD235a$^{neg}$/CD19$^{pos}$/HLA-E-VL9$^{double-pos}$/HLA-E-RL9HIV$^{neg}$/HLA-E-RL9SIV-$^{neg}$ subset as single cells in 96-well plates.

**PCR amplification of human antibody genes**. The $V_HD_HJ_H$ and $V_LJ_L$ genes were amplified by reverse transcription polymerase chain reaction (RT-PCR) from the flow cytometry-sorted single B cells using the methods as described previously[72,73] with modification. The PCR-amplified genes were then purified and sequenced with 10 μM forward and reverse primers. Sequences were analyzed by using the human library in Clonalyst for the VDJ arrangements of the immunoglobulin *IGHV*, *IGKV*, and *IGLV* sequences and mutation frequencies[41]. Clonal relatedness of $V_HD_HJ_H$ and $V_LJ_L$ sequences was determined as previously described[74].

**Expression of $V_HD_HJ_H$ and $V_LJ_L$ as full-length IgG recombinant mAbs**. Transient transfection of recombinant mAbs was performed as previously described[73]. Briefly, purified PCR products were used for overlapping PCR to generate linear human antibody expression cassettes. The expression cassettes were transfected into 293i cells using ExpiFectamine (Thermo Fisher Scientific, Catalog# A14525). The supernatant samples containing recombinant antibodies were used for cell surface staining and HTS assay to measure the binding reactivities.

The selected human antibody genes were then synthesized and cloned (GenScript) in a human IgG1 backbone with 4A mutations[75]. Recombinant IgG mAbs were then produced in HEK293i suspension cells by transfection with ExpiFectamine and purified using Protein A resin. The purified mAbs were run in SDS-PAGE for Coomassie blue staining and western blot. Antibodies with aggregation were further purified in AKTA FPLC system using a Superdex 200 size-exclusion column.

**Surface plasmon resonance (SPR)**. Surface plasmon resonance assays were performed on a BIAcore 3000 instrument, and data analysis was performed with BIAevaluation 3.0 software as previously described[76]. Purified mAbs flowed over CM5 sensor chips at concentrations of 100 μg/ml, and antibody binding was monitored in real-time at 25 °C with a continuous flow of PBS at 30 μl/min. For SPR affinity measurements, antibody binding to HLA-E-VL9 complex protein was performed using a BIAcore S200 instrument (Cytiva, formerly GE Healthcare, DHVI BIA Core Facility, Durham, NC) in HBS-EP+ 1x running buffer. The antibodies were first captured onto CM5 sensor chip to a level of ~9000 RU. The HLA-E-VL9 soluble proteins were injected over the captured antibodies at a flow rate of 30 μl/min. After dissociation, the antibodies were regenerated using a 30 second pulse of Glycine pH 2.0. Results were analyzed using the Biacore S200 Evaluation software (Cytiva). Subsequent curve fitting analyses were performed using a 1:1 Langmuir model with a local Rmax. The reported binding curves are representative of two data sets.

**Enzyme-linked immunosorbent assay (ELISA)**. Direct binding ELISAs were conducted in 384-well ELISA plates coated with 2 μg/ml of C-trap-stabilized HLA-E-VL9, C-trap-stabilized HLA-E-RL9HIV or C-trap-stabilized HLA-E-RL9SIV in 0.1 M sodium bicarbonate overnight at 4 °C. Plates were washed with PBS + 0.05% Tween 20 and blocked with 3% BSA in PBS at room temperature for 1 h. MAb samples were incubated for 1 h in threefold serial dilutions starting at 100 μg/ml, followed by washing with PBS-0.05% Tween 20. HRP-conjugated goat anti-human IgG secondary Ab (SouthernBiotech, catalog# 2040-05) was diluted to 1:10,000 in 1% PBS-0.05% Tween 20 and incubated at room temperature for 1 h. For sandwich ELISA, 384-well ELISA plates were coated with HLA-E-VL9 antibodies in a threefold dilution starting from 100 μg/ml in 0.1 M sodium bicarbonate overnight at 4 °C. Plates were washed with PBS + 0.05% Tween 20 and blocked with 3% BSA in PBS at room temperature for 1 h. C-trap-stabilized HLA-E-VL9, C-trap-stabilized HLA-E-RL9HIV, C-trap-stabilized HLA-E-RL9SIV, or diluent control were then added at 2 μg/ml and incubated at room temperature for 1 h. After washing, HRP-conjugated anti-human β2M antibody (Biolegend, catalog# 280303) were added at 0.2 μg/ml and incubated at room temperature for 1 h. These plates were washed for four times and developed with tetramethylbenzidine substrate (SureBlue Reserve). The reaction was stopped with 1 M HCl, and optical density at 450 nm ($OD_{450}$) was determined.

**Antibody poly-reactivity assays**. All mAbs isolated from mice and human were tested for ELISA binding to nine autoantigens—Sjogren's syndrome antigen A (SSA), Sjogren's syndrome antigen (SSB), Smith antigen (Sm), ribonucleoprotein (RNP), scleroderma 70 (Scl-70), Jo-1 antigen, double-stranded DNA (dsDNA), centromere B (Cent B), and histone as previously described[77,78]. Indirect immunofluorescence assay of mAbs binding to HEp-2 cells (Inverness Medical Professional Diagnostics, Princeton, NJ) was performed as previously described[78,79]. MAbs 2F5[80] and 17B[81] were used as positive and negative controls, respectively. All antibodies were screened in two independent experiments.

**Negative-stain electron microscopy of IgM antibodies**. FPLC purified IgM antibodies were diluted to 0.08 mg/ml in HEPES-buffered saline (pH 7.4) + 5% glycerol, and stained with 2% uranyl formate. Images were obtained with a Philips EM420 electron microscope at 82,000 magnification and processed in Relion 3.0.

**Peptide-pulsing in K562-E cells**. K562-E cells and K562-E/UL49.5 cells were resuspended with fresh IMDM media with 10% FBS at $2 \times 10^6$ cells/ml. Peptides were added into cell suspension at a final concentration of 100 μM. The cell/peptide mixtures were incubated at 26 °C with 5% $CO_2$ for 20–22 h and were transferred to 37 °C for 2 h with 5% $CO_2$ before use. In the following mAb staining experiment, medium with 100 μM peptides was used to maintain peptide concentration.

**NK cell cytotoxicity assay**. NK cell cytotoxicity was measured by $^{51}Cr$ release assay. A NKG2A-positive, CD16/CD32/CD64-negative NK-92 cells were used as effector cells in our study. Transfected or untransfected 293T cells were used as target cells. Target cells were counted, washed, resuspended in R10 at $1 \times 10^7$ cell/ml, and labeled with $Na_2{}^{51}CrO_4$ at 250 μCi/ml for 2 h at 37 °C. After washing three times using R10, cells were mixed with the testing antibody and effector cells in a final effector to target (E:T) ratio of 20:1 and 6:1 in triplicate wells in a flexible 96-well round bottom plates (PerkinElmer, Catalog# 1450-401). The plates were inserted in flexible 96-well plate cassettes (PerkinElmer, Catalog# 1450-101), sealed and incubated at 37 °C for 4 h. After the incubation, cells were pelleted by centrifugation, and from the top of the well, add 25 μl of supernatant to a rigid 96-well isoplates (PerkinElmer, Catalog#1450-514) containing 150 μl of Ultima Gold LSC Cocktail (Sigma, Catalog# L8286). The plates were inserted in rigid 96-well plate cassettes (PerkinElmer, Catalog# 1450-105), sealed and counted on Perkin Elmer Microbeta Triux 1450 counter. $^{51}Cr$ labeled target cells without effector cells were set as a spontaneous release control, and $^{51}Cr$ labeled target cells mixed with detergent (2% Triton X-100) were used as a maximum release control. The percentages of specific lysis were calculated with the formulation: The Percentages of Specific Lysis ($^{51}Cr$ Release%) = [(Experimental Release – Spontaneous Release)/(Maximum Release – Spontaneous Release)] × 100.

**Development and screening of scFv libraries on the surface of yeast**. A library was built that contained ~1.1 million 3H4 scFv variants with amino acid diversity at sites that were determined by structural analysis to interact with HLA-E-VL9. Seventeen residues (Supplementary Fig. 3) located in the CDR loops of 3H4 were randomized in groups of four based on their proximity and all the possible combinations of amino acids were sampled at these sites. Library DNA was synthesized on a BioXP 3250 (Codex) system and amplified with High Fidelity Phusion polymerase (New England Biolabs). PCR products were gel extracted (Qiagen Gel Extraction kit) to select full-length genes as per the manufacturer's protocol. 3H4 scFv variants were displayed in library format on the surface of yeast as previously described[82,83]. Briefly, *S. cerevisiae* EBY100 cells were transformed by electroporation with a 3:1 ratio of 12 μg scFv library DNA and 4 μg pCTcon2 plasmid digested with BamHI, SalI, NheI (New England Biolabs). The size of the transformed library, determined by serial dilution on selective plates, was $5 \times 10^7$ individual colonies. Yeast Libraries were grown in SDCAA media (Teknova) supplemented with pen-strep at 30 °C and 225 rpm. Eighty percent of the sequences recovered from the transformed libraries were confirmed to contain full length, in-frame genes by Sanger sequencing (Genewiz). scFv expression on the surface of yeast was induced by culturing the libraries in SGCAA (Teknova) media at a density of $1 \times 10^7$ cells/ml for 24–36 h. Cells were washed twice in ice-cold PBSA (0.01 M sodium phosphate, pH 7.4, 0.137 M sodium chloride, 1 g/L bovine serum albumin) and labeled with APC-conjugated HLA-E-VL9 tetramer and 1:100 anti-c-myc:FITC (ICL) and incubated for 1 h at 4 °C. Initial selection was conducted with 50 mg/ml labeling concentration of HLA-E-VL9 tetramer; the second round of selection was done at 0.6 mg/ml tetramer. Cells were washed twice with PBSA after incubation with the fluorescently labeled probes and sorted on a BD FACS-DiVa. Double-positive cells for APC and FITC were collected and expanded in SDCAA media supplemented with pen-strep before successive rounds of enrichment. FACS data was analyzed with Flowjo_v10.7 software (Becton, Dickinson & Company). All clones selected by FACS were expanded, and their DNA was extracted (Zymo Research) for analysis by Sanger sequencing (Genewiz). scFv encoding plasmids were recovered from yeast cultures by yeast miniprep with the Zymoprep yeast plasmid miniprep II kit (Zymo Research). Isolated DNA was transformed into NEB5α strain of *E. coli* (New England Biolabs) and the DNA of individual bacterial colonies was isolated (Wizard Plus SV Minipreps, Promega) and analyzed by Sanger sequencing (Genewiz).

**Statistics analysis and reproducibility**. Data were plotted using Prism GraphPad 8.0 or visualized using the ComplexHeatmap R package. SAS 9.4 (SAS Institute, Cary, NC) was used to perform the statistical analysis with a $P$-value < 0.05 considered significant. For $^{51}Cr$ release assays, mixed effects models were used to make comparisons of antibody to control using a random intercept for the triplicates run within each experiment and fixed effects of E:T ratio, type (antibody or control), and the interaction of E:T ratio by type. For human antibody gene usage analysis, chi-square test of independence was used to compare differences between groups. All attempts at replication were successful in 3–6 independent experiments, as indicated in the figure legends.

**Reporting summary**. Further information on research design is available in the Nature Research Reporting Summary linked to this article.

## Data availability

All data generated or analyzed during this study are included in this published article and Supplementary Data, or are available from the corresponding author upon reasonable request. The source data underlying the graphs and charts in the figure are provided in Supplementary Data 4. The 3H4-HLA-E-VL9 co-complex structure determined in this study has the PDB accession code, 7BH8.

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

## Acknowledgements

We thank Duke Human Vaccine Institute (DHVI) programs and finance staff for project oversight and the contributions of technical staff at the DHVI, including Jordan Cocchiaro, Kelly Soderberg, Ahmed Yousef Abuahmad, Yunfei Wang, Giovanna Hernandez, Esther Lee, Paige Power, Aja Sanzone, Brenna Harrington, Andrew Foulger, Amanda Newman, Cindy Bowman, Grace Stevens, Laura Sutherland, Margaret Deyton, Victoria Gee-Lai, Tarra Von Holle, Thad Gurley, Madison Berry, Kara Anasti, Katayoun Mansouri, Erika Dunford, and Dawn Jones Marshall. We thank the DHVI Flow Cytometry Core and Duke Cancer Institute Flow Cytometry Shared Resource (FCSR) for technical assistance. This study was funded by the Collaboratory of AIDS Researchers for Eradication (CARE; UM1-AI-126619 and UM1-AI-164567) (B.F.H.) and the Bill and Melinda Gates Foundation OPP1108533 (A.J.M.). P.B and A.J.M. are Jenner Institute Investigators.

## Author contributions

D.L. isolated and characterized antibodies, performed $^{51}$Cr release assay, and analyzed data. S.B. made single-chain trimer constructs, performed epitope mapping experiments and analyzed data. G.M.G., L.W. and M.Q. prepared the antigens and HLA-E tetramers. L.W., G.M.G., D.R. and K.H performed structural experiments with oversight from E.Y.J.O.S. and M.L.A. performed antibody affinity optimization work. D.W.C. helped with flow cytometry sorting set-up. R.S., R.P. and M.B. performed hybridoma experiments, ELISA assays and help with $^{51}$Cr release assays. R.J.E. oversaw negative stain EM. M.A. supervised and interpreted the SPR experiments. K.W. oversaw and performed antibody gene sequence analysis. Z.M. and M.B. helped with antibody isolation. Y.W. and W.R. provided statistical analysis. K.O.S. oversaw antigen production. G.F. and P.B. provided advice on NK cell assays and contributed to study design and data interpretation. B.F.H. and A.J.M. conceived, designed, coordinated the study. D.L. and B.F.H. wrote the manuscript, which was edited by L.W., S.B., O.S., M.L.A., P.B., G.M.G., and A.J.M., and reviewed by all authors.

## Competing interests

The authors declare the following competing interests: D.L., S.B., M.L.A., G.M.G., A.J.M. and B.F.H. have patents submitted on antibodies and select methods in this paper. All other authors declare no competing interests.
