## [Transparent Peer Review File · Communications Biology]

Reviewers' comments:

Reviewer #1 (Remarks to the Author):

With the aim of generating antibodies to the HIV-1 Gag peptide, the authors inadvertently generated monoclonal antibodies specific for HLA-E-VL9 peptide. The 3H4 clone (IgM) binds to both HLA-E $\alpha 1$ and $\alpha 2$ domains and amino-terminal of VL9 peptide (residues 1-4), preventing NKG2A/CD94 engagement due to steric hindrance. The four 3H4-derived residues (Y97, S100, S100A and Y100B) that interact with both the HLA-E heavy chain and VL9 peptide are germline-encoded. 3H4 binding to NKG2A/CD94 enhanced NK92 cytotoxicity against HLA-E-VL9 293T cells in vitro. The authors then generated a recombinant 3H4 in a human IgG1 backbone which lacked cytotoxicity activity. However, affinity-optimised variants G3v and G6v had comparable cytotoxicity to IgM multimers. Lastly, HLA-E-VL9-specific antibodies were found in CMV- human donors. These antibodies were predominantly found to be expressed by naïve B cells with minimal somatic mutation.

The manuscript is technically sound and well-written, with detailed structural data and antibody characterisation. The generation and characterization of HLA-E-VL9-specific antibodies and their identification in the natural repertoire of human B cells is a novel, albeit relatively minor advance, with their relevance in human health or disease not being particularly clear. The functional importance of HLA-E-VL9 antibodies which bypass NKG2A/CD94-mediated inhibition of NK cell cytotoxicity is unclear, and may represent an escape from tolerance which is not particularly damaging and thus remains in the repertoire. The potential for clinical use as an NK checkpoint inhibitor is an interesting point mentioned in the discussion, although the 3H4 clone does not appear to have any advantage over the existing Z199 anti-NKG2A antibody in the data presented in Figure 3. Despite this, the data presented convincingly support the claims made in the paper, and will be of interest to some in the field.

Reviewer #2 (Remarks to the Author):

In this study, Li, Brackenridge, Walters, et al. isolated and characterized mouse and human antibodies reactive with HLA-E-VL9 peptide complexes. Mouse antibodies were derived from mice immunized with HLA-E-unrelated peptide. These antibodies mostly derived from the naïve IgM B cell BCR repertoire, somatic mutations were minimal, and their affinities for HLA-E-VL9 were low. The authors propose that the lack of class-switching in HLA-E-VL9-specific antibodies may reflect self-tolerance of CD4 T cells and a lack of T cell help for affinity maturation of these antibodies. Human antibodies were derived from CMV seronegative human blood donors. The human antibodies that bound to human HLA-E-VL9 were auto-antibodies and were also near germline.

This is a novel and very interesting study that highlights the potential generation of anti-HLA-E antibodies with a peptide specificity in the settings of autoimmunity or HLA-E-unrelated peptide immunizations. The quality of the work is outstanding, I am impressed with the detailed analysis of the antibodies, which include the structural analysis of the HLA-E-VL9-3H4 Fab co-complex to identify the key contacts of the germline-encoded residues in the CDR-H3 region with HLA-E and the VL9 peptide residues. I congratulate the authors for this excellent study

Reviewer #1 (Remarks to the Author):

With the aim of generating antibodies to the HIV-1 Gag peptide, the authors inadvertently generated monoclonal antibodies specific for HLA-E-VL9 peptide. The 3H4 clone (IgM) binds to both HLA-E $\alpha 1$ and $\alpha 2$ domains and amino-terminal of VL9 peptide (residues 1-4), preventing NKG2A/CD94 engagement due to steric hindrance. The four 3H4-derived residues (Y97, S100, S100A and Y100B) that interact with both the HLA-E heavy chain and VL9 peptide are germline-encoded. 3H4 binding to NKG2A/CD94 enhanced NK92 cytotoxicity against HLA-E-VL9 293T cells in vitro. The authors then generated a recombinant 3H4 in a human IgG1 backbone which lacked cytotoxicity activity. However, affinity-optimised variants G3v and G6v had comparable cytotoxicity to IgM multimers. Lastly, HLA-E-VL9-specific antibodies were found in CMV- human donors. These antibodies were predominantly found to be expressed by naïve B cells with minimal somatic mutation.

The manuscript is technically sound and well-written, with detailed structural data and antibody characterisation. The generation and characterization of HLA-E-VL9-specific antibodies and their identification in the natural repertoire of human B cells is a novel, albeit relatively minor advance, with their relevance in human health or disease not being particularly clear. The functional importance of HLA-E-VL9 antibodies which bypass NKG2A/CD94-mediated inhibition of NK cell cytotoxicity is unclear, and may represent an escape from tolerance which is not particularly damaging and thus remains in the repertoire. The potential for clinical use as an NK checkpoint inhibitor is an interesting point mentioned in the discussion, although the 3H4 clone does not appear to have any advantage over the existing Z199 anti-NKG2A antibody in the data presented in Figure 3. Despite this, the data presented convincingly support the claims made in the paper, and will be of interest to some in the field.

Answer: We thank the reviewer for this supportive comment. We have now added discussion text regarding to the potential role of the endogenous anti-HLA-E-VL9-antibodies in Line 327-332.

Reviewer #2 (Remarks to the Author):

In this study, Li, Brackenridge, Walters, et al. isolated and characterized mouse and human antibodies reactive with HLA-E-VL9 peptide complexes. Mouse antibodies were derived from mice immunized with HLA-E-unrelated peptide. These antibodies mostly derived from the naïve IgM B cell BCR repertoire, somatic mutations were minimal, and their affinities for HLA-E-VL9 were low. The authors propose that the lack of class-switching in HLA-E-VL9-specific antibodies may reflect self-tolerance of CD4 T cells and a lack of T cell help for affinity maturation of these antibodies. Human antibodies were derived from CMV seronegative human blood donors. The human antibodies that bound to human HLA-E-VL9 were auto-antibodies and were also near germline.

This is a novel and very interesting study that highlights the potential generation of anti-HLA-E antibodies with a peptide specificity in the settings of autoimmunity or HLA-E-unrelated peptide immunizations. The quality of the work is outstanding, I am impressed with the detailed analysis of the antibodies, which include the structural analysis of the HLA-E-VL9-3H4 Fab co-complex

to identify the key contacts of the germline-encoded residues in the CDR-H3 region with HLA-E and the VL9 peptide residues. I congratulate the authors for this excellent study

Answer: We appreciate the supportive comments from the reviewer.